# Best of Both Worlds: Regret Minimization versus Minimax Play

**Adrian Müller** [* 1]   **Jon Schneider** [* 2]   **Stratis Skoulakis** [* 3]   **Luca Viano** [* 4]   **Volkan Cevher** [4]

## Abstract

In this paper, we investigate the existence of online learning algorithms with bandit feedback that simultaneously guarantee $O(1)$ regret compared to a given comparator strategy, and $\tilde{O}(\sqrt{T})$ regret compared to any fixed strategy, where $T$ is the number of rounds. We provide the first affirmative answer to this question whenever the comparator strategy supports every action. In the context of zero-sum games with min-max value zero, both in normal- and extensive form, we show that our results allow us to guarantee to risk at most $O(1)$ loss while being able to gain $\Omega(T)$ from exploitable opponents, thereby combining the benefits of both no-regret algorithms and minimax play.

## 1. Introduction

Two-player zero-sum games form one of the most fundamental classes studied in game theory, capturing direct competition between two opposing agents. In a zero-sum game, Alice and Bob choose mixed strategies $\mu \in \mathcal{P}$ and $\nu \in \mathcal{P}'$, respectively, from some strategy polytopes $\mathcal{P}$ and $\mathcal{P}'$. Their expected payoffs are specified by a function $V$. Alice aims to minimize $V(\mu, \nu)$, whereas Bob aims to maximize it. This definition subsumes the classical *normal-form zero-sum games* (Von Neumann & Morgenstern, 2007) like Rock-Paper-Scissors, as well as the more complex *extensive-form zero-sum games* (Osborne & Rubinstein, 1994), such as Heads-up Poker. A zero-sum game is called *fair* if its min-max value is zero, meaning that $\min_{\mu \in \mathcal{P}} \max_{\nu \in \mathcal{P}'} V(\mu, \nu) = 0$. This models the fact that none of the players has a strategic advantage due to the structure of the game. For instance, a game is always fair if it is symmetric, i.e. when $\mathcal{P} = \mathcal{P}'$ and $V(\mu, \nu) = -V(\nu, \mu)$, as is the case for many games of interest. Now suppose Alice repeatedly plays a fair zero-sum game against her unknown opponent Bob for $T$ consecutive rounds. In each round, she chooses her next strategy based on all her previous observations, and Bob does likewise. Both players then receive their respective costs in this round prior to moving to the next round.

To minimize her cumulative cost, Alice could compute an equilibrium strategy and simply play it in every round (*minimax play* (Von Neumann & Morgenstern, 2007)). This way, she would be guaranteed to never lose anything to Bob in expectation. However, she might also not win anything from Bob even if he plays suboptimal (non-equilibrium) strategies. A classic example of this dilemma is Rock-Paper-Scissors, for which the min-max strategy is the uniform strategy, which wins zero even from an opponent that always plays Rock. Alternatively, Alice could run a learning algorithm (*regret minimization* (Cesa-Bianchi & Lugosi, 2006)). This way, her average cost would approach the one of the best strategy in hindsight, allowing her to exploit such opponents. However, by running such an algorithm she would have to deviate from the equilibrium strategy, thereby risking incurring a significant amount of costs during learning. More formally, there are two popular lines of thought on how Alice could minimize her overall cost over the $T$ rounds of play:

**1) Min-Max Equilibrium:** In every round $t$, Alice simply selects the min-max strategy $\mu^t = \mu^\star \in \arg\min_{\mu \in \mathcal{P}} \max_{\nu \in \mathcal{P}'} V(\mu, \nu)$. She then loses at most $V^\star := \min_{\mu \in \mathcal{P}} \max_{\nu \in \mathcal{P}'} V(\mu, \nu)$ units to Bob. For fair zero-sum games, we have $V^\star = 0$, meaning that she will not lose anything in expectation. However, for example in normal-form games, she also never wins any units if $\mu^\star$ is full-support (Braggion et al., 2020), and even otherwise may not win anything (Section 5). In summary:

*Alice is guaranteed not to lose anything, but might not win anything even if Bob plays poorly.*

**2) Regret Minimization:** Alice selects $\mu^t \in \mathcal{P}$ according to a *no-regret algorithm*. Then she can guarantee that, no matter Bob's strategies $\nu^1, \dots, \nu^T \in \mathcal{P}'$, the *regret* compared to any fixed strategy $\mu$ satisfies

$$\sum_{t=1}^{T} V(\mu^t, \nu^t) - \sum_{t=1}^{T} V(\mu, \nu^t) \leq O(\sqrt{T}).$$

In fair zero-sum games, plugging in the equilibrium $\mu = \mu^\star$, we have $V(\mu, \nu^t) \leq V^\star = 0$. This means that the above regret guarantee ensures that Alice might lose at most

[1]ETH Zürich [2]Google Research [3]Aarhus University [4]EPFL. Correspondence to: Adrian Müller <admuell@ethz.ch>.

*Proceedings of the 42$^{nd}$ International Conference on Machine Learning*, Vancouver, Canada. PMLR 267, 2025. Copyright 2025 by the author(s).

$O(\sqrt{T})$ units to Bob, which can be a significant amount. Indeed, one expects there are cases where she does (Section 5) since there is a matching regret lower bound. However, if Bob plays sub-optimally, it may be the case that $\min_{\mu \in \mathcal{P}} \sum_{t=1}^{T} V(\mu, \nu^t) = -\Theta(T)$, meaning that Alice wins $\Theta(T)$ units. As a result:

> *Alice risks losing $O(\sqrt{T})$ units, but can win up to $\Theta(T)$ if Bob plays sub-optimally.*

Whether Alice will choose to play 1) a min-max equilibrium or 2) according to a no-regret algorithm depends on how *risk-averse* Alice is — how willing Alice is to risk $O(\sqrt{T})$ units in the hope of winning $\Theta(T)$. This naturally raises the question of whether we can have the best of both worlds:

**Question 1.** *In a fair zero-sum game, can Alice risk losing at most $O(1)$ units, but still be able to win up to $\Theta(T)$ if Bob plays sub-optimally?*

In this paper, we answer this question in the affirmative by resolving the following fairly *more general question* from online learning with adversarial linear costs. We explain the reduction in Section 2.

**Question 2.** *Is it possible to guarantee $O(1)$ regret compared to a specific strategy while maintaining $\tilde{O}(\sqrt{T})$ regret compared to any fixed strategy?*

Question 2 is known to admit a relatively simple positive answer in the so-called full-information case (Section 1.1). Crucially, in this work we are interested in the *bandit feedback* setting, modeling the fact that Alice only observes the realized cost and not the cost for all actions she could have taken instead. We formalize this learning goal in Sections 3.1 and 4.1.

We present our results in the context of fair zero-sum games. However, they hold far beyond fair, zero-sum, or even two-player games (Question 2): for any sufficiently explorative comparator strategy, one can guarantee constant regret compared to it while still having rate-optimal regret compared to any fixed strategy $\mu$, even under bandit feedback. Our results may thus be of independent interest to the online learning community, as we discuss in Section 1.1.

**Contributions.** Our main contributions are the following:

- We first devise an algorithm for normal-form games (NFGs) under bandit feedback that interpolates between playing the min-max equilibrium and no-regret learning. We prove that if the min-max equilibrium is supported on the whole action space[1], then our algorithm indeed satisfies the desiderata of our main question (Section 3.2).

---

[1]This assumption is also necessary, but can easily be relaxed, at the cost of slightly weaker guarantees on when Alice can take advantage of sub-optimal play by Bob. See Remark 3.1.

To the best of our knowledge, this is the first result of its kind under bandit feedback.

- We complement this regret guarantee with a lower bound for NFGs, showing that the regret bound cannot be improved significantly (Section 3.3). This illustrates that our algorithm is close to optimally exploiting weak strategies, as desired.

- We then transfer our insights to the more challenging framework of extensive-form games (EFGs). This is specifically relevant since in stateful games, it is essential to consider bandit feedback. By proposing a corresponding algorithm for EFGs, we show that even in such interactive games with imperfect information, we can answer our main question in the affirmative (Section 4.2). We generalize our lower bound to this setting, too (Section 4.3).

Finally, we numerically evaluate our algorithm in simple EFG environments (Section 5), showing that our results are not merely of theoretical interest. Indeed, our findings confirm our theoretical insights and demonstrate strong results even when the min-max equilibrium is not full-support.

## 1.1. Related Work

In online learning under *full information feedback*, it is known that one can achieve constant regret against a certain comparator strategy while maintaining the near-optimal worst-case regret guarantee as desired in Question 2 (Hutter et al., 2005; Even-Dar et al., 2008; Kapralov & Panigrahy, 2011; Koolen, 2013; Sani et al., 2014; Orabona & Pál, 2016; Cutkosky & Orabona, 2018; Orabona, 2019), one notable example being the Phased Aggression template of Even-Dar et al. (2008). This allows us to directly answer Question 1 affirmatively for NFGs if full information is available, via the reduction in Section 2. While this reduction is direct, we are not aware of any prior work making this connection, even under full-information feedback.

In stark contrast, under *bandit feedback*, Lattimore (2015) showed that in multi-armed bandits, $O(1)$ regret compared to a single comparator action (i.e. a deterministic strategy) implies a worst-case regret of $\Omega(AT)$ compared to some other action. This rules out a positive answer to our Question 2 if the comparator strategy is arbitrary. We show that, perhaps surprisingly, it is possible to circumvent this lower bound under the minimal possible assumption that the comparator strategy plays each action with non-zero probability $\delta > 0$ (while maintaining the optimal order of $\sqrt{T}$ regret).

Similar to our motivation, Ganzfried & Sandholm (2015) consider *Safe Opponent Exploitation* as deviating from the min-max strategy while ensuring at most the cost of the min-max value. Different from our work, their algorithms rely on best-responding to some opponent model whenever the algorithm has accumulated enough utility to risk losing it again. While the authors provide safety guarantees, they do

not provide any theoretical exploitation guarantee. In contrast, our algorithm has provably vanishing regret compared to the best static response against the opponent.

Regarding the extension of our results to EFGs, we leverage relatively recent theoretical advancements regarding online mirror descent in EFGs, most notably Kozuno et al. (2021); Bai et al. (2022). Finally, we refer to Appendix A for an extended discussion of related work.

## 2. Preliminaries

In this section, we introduce the relevant notation and explain how Question 2 answers Question 1.

**Notation.** As usual, $O$-notation expresses asymptotic behavior, and $\tilde{O}$-notation hides poly-logarithmic factors. We denote the $n$-dimensional simplex by $\Delta_n$ and define $[n] := \{1, \ldots, n\}$. Moreover, $e_i$ denotes the $i$-th the standard basis vector of $\mathbb{R}^n$, and $\langle \cdot, \cdot \rangle$ the Euclidean inner product. Finally, we write $\mathbb{1}_E$ for the indicator function of an event $E$.

**(Safe) Online Linear Minimization.** In Protocol 1, we introduce the framework of *online linear minimization* (Hazan, 2019, OLM) with adversarial costs. In addition to this standard framework, Alice receives a special *comparator strategy* $\mu^c \in \mathcal{P}$ she considers "safe". The motivation for this is that we can later choose $\mu^c$ to be a min-max equilibrium $\mu^\star$, which is safe in the sense of guaranteeing zero expected loss in fair zero-sum games. Alice would like to be essentially at least as good as this comparator strategy.

---

**Protocol 1** (Safe) Online Linear Minimization

---

**Require:** Special comparator $\mu^c \in \mathcal{P}$.
    **for** round $t = 1, \ldots, T$ **do**
        **Alice** chooses her next $\mu^t \in \mathcal{P}$.
        **Bob** chooses the cost vector $c^t$.
        **Alice** suffers expected cost $\langle \mu^t, c^t \rangle$.
**Goal:** $\mathcal{R}(\mu^c) \leq O(1)$ and $\max_{\mu \in \mathcal{P}} \mathcal{R}(\mu) \leq \tilde{O}(\sqrt{T})$.

---

We define Alice's *expected regret compared to a strategy* $\mu \in \mathcal{P}$ by

$$\mathcal{R}(\mu) := \sum_{t=1}^{T} \mathbb{E}\left[\langle \mu^t - \mu, c^t \rangle\right].$$

The *expected regret* $\max_\mu \mathcal{R}(\mu) = \sum_{t=1}^{T} \mathbb{E}\left[\langle \mu^t, c^t \rangle\right] - \min_\mu \sum_{t=1}^{T} \mathbb{E}\left[\langle \mu, c^t \rangle\right]$ then measures the regret compared to the best fixed strategy $\mu$ in hindsight. Under *safe OLM* (Question 2), we understand the problem of simultaneously guaranteeing

$$\mathcal{R}(\mu^c) \leq O(1), \quad \text{and} \quad \max_{\mu \in \mathcal{P}} \mathcal{R}(\mu) \leq \tilde{O}(\sqrt{T}). \quad \text{(OLM)}$$

**Question 2 Answers Question 1.** Now suppose Alice was able to guarantee (OLM). As we explain in Sections 3.1 and 4.1, both for NFGs and EFGs, we can write the expected cost in round $t$ as a linear function of the strategy, i.e.

$$\mathbb{E}\left[V(\mu, \nu^t)\right] = \mathbb{E}\left[\langle \mu, c^t \rangle\right]$$

for some cost vector $c^t$. Alice can now set $\mu^c = \mu^\star = \arg\min_\mu \max_\nu V(\mu, \nu)$ to be a min-max equilibrium. Since $V(\mu^c, \nu) \leq V^\star = 0$ for fair zero-sum games, the first part of (OLM) implies

$$\sum_{t=1}^{T} \mathbb{E}\left[V(\mu^t, \nu^t)\right] \leq \sum_{t=1}^{T} \mathbb{E}\left[V(\mu^c, \nu^t)\right] + O(1) \leq O(1),$$

no matter Bob's play. Alice will thus lose at most a constant amount in expectation. Furthermore, if (for example) Bob plays a fixed strategy $\nu^t = \nu \in \mathcal{P}'$ that is suboptimal in the sense that $\min_\mu V(\mu, \nu) = -c < 0$, then the second part in (OLM) shows

$$\sum_{t=1}^{T} \mathbb{E}\left[V(\mu^t, \nu^t)\right] \leq \min_\mu \sum_{t=1}^{T} V(\mu, \nu) + \tilde{O}(\sqrt{T}) \leq -\Theta(T),$$

and Alice will linearly exploit Bob.[2] We will thus state our results in terms of safe OLM, keeping in mind that the above reduction will automatically answer our initial Question 1.

## 3. Normal-Form Games

Suppose Alice and Bob repeatedly play a *normal-form zero-sum game* for $T$ rounds, which means the following. In each round $t$, they simultaneously submit actions $a^t \in [A]$, $b^t \in [B]$ by sampling from mixed strategies $\mu^t \in \Delta_A$, $\nu^t \in \Delta_B$, respectively. Alice receives cost $U_{a^t, b^t} = \langle e_{a^t}, U e_{b^t} \rangle$ and Bob receives cost $-U_{a^t, b^t}$, for some fixed cost matrix $U \in \mathbb{R}^{A \times B}$ with entries in $[0, 1]$. Alice's expected cost given $\mu^t, \nu^t$ is $V(\mu^t, \nu^t) := \mathbb{E}_{a \sim \mu^t, b \sim \nu^t}[U_{a,b}] = \langle \mu^t, U \nu^t \rangle$. We consider *bandit feedback*, meaning that Alice only observes her cost $U_{a^t, b^t}$ and not the cost of actions she could have taken instead.

### 3.1. From NFGs to Online Linear Minimization
By defining Alice's *cost function* as

$$c^t := U e_{b^t} \in \mathbb{R}^A,$$

we see that Alice's expected cost is $\mathbb{E}\left[V(\mu^t, \nu^t)\right] = \mathbb{E}[U_{a^t, b^t}] = \mathbb{E}[\langle \mu^t, c^t \rangle]$, as $a^t \sim \mu^t$. We are thus in the setting of OLM (Protocol 1) over $\mathcal{P} = \Delta_A$. Notably, Alice does not observe the full cost function $c^t$ but only its entry $c^t(a^t) = U_{a^t, b^t}$ at the chosen action (bandit feedback). We

---

[2]More generally, Bob is *exploitable* in this sense if he plays an oblivious sequence of strategies $\nu^t$ with $\min_\mu \sum_t V(\mu, \nu^t) = -\Theta(T)$. We briefly discuss the adaptive case in Appendix A.

formally consider Protocol 2 for *any* adversarially picked cost functions $c^t$. From Section 2 we know that it is now sufficient to set $\mu^c = \mu^\star$ and guarantee (OLM).

---

**Protocol 2** Bandit Feedback over the Simplex (NFGs)

---

**Require:** Special comparator $\mu^c \in \Delta_A$.
   **for** round $t = 1, \ldots, T$ **do**
      **Alice** chooses her next action $a^t \sim \mu^t \in \Delta_A$.
      **Bob** chooses costs $c^t \in \mathbb{R}^A$.     ▷ NFG: $c^t = U e_{b^t}$
      **Alice** suffers and observes cost $c^t(a^t)$. ▷ NFG: $U_{a^t, b^t}$

---

### 3.2. Upper Bound

Our first main result shows that if the special comparator strategy lies in the interior of the simplex, we are able to guarantee constant regret to it while maintaining low regret to any strategy at the optimal rate in $T$. Note that the result concerns the general Protocol 2 and thus covers *any* NFG, which need not be fair or zero-sum (or even two-player).

**Theorem 3.1.** *Let* $\delta \in (0, 1/A]$. *Consider any mixed strategy* $\mu^c \in \Delta_A$ *such that* $\mu^c(a) \geq \delta$ *for all* $a \in [A]$. *Under bandit feedback (Protocol 2), for any sequence of* $c^t \in [0, 1]^A$, *Algorithm 1 achieves*

$$\mathcal{R}(\mu^c) \leq 1, \quad and \quad \max_{\mu \in \Delta_A} \mathcal{R}(\mu) \leq \tilde{O}\left(\delta^{-1}\sqrt{T}\right).$$

Now consider any *zero-sum* NFG with min-max value $V^\star$. If the min-max strategy $\mu^\star$ is full-support, then Alice can run Algorithm 1. The above theorem and the reduction from Section 2 guarantee that in expectation: Alice will lose at most $V^\star T + 1$ units while winning $\Omega(T)$ if Bob plays (oblivious) strategies that are linearly exploitable. In particular, if the game is fair ($V^\star = 0$), Alice will lose at most 1 unit in expectation. The latter is guaranteed for instance if the zero-sum NFG is symmetric (i.e. $\mathcal{A} = \mathcal{B}$ and $U = -U^T$).

Lattimore (2015)'s result implies that the assumption on $\mu^c$ is also necessary. In addition, we show in Theorem 3.2 that a multiplicative dependence on $\delta^{-1}$ is unavoidable. We remark that min-max strategies $\mu^\star$ of various zero-sum games are $\delta$-bounded away from zero. For example in Rock-Paper-Scissors $\mu^\star = (1/3, 1/3, 1/3)$. More importantly, even when this is not the case, we remark the following.

**Remark 3.1.** *Alice can apply the result even in zero-sum games with min-max strategies* $\mu^\star \in \Delta_A$ *that are not full-support. Indeed, she can consider the subset of actions* $\mathcal{A}' := \{a \in [A]: \mu^\star(a) > 0\}$. *Then, our algorithm run on* $\mathcal{A}'$ *still guarantees* $\mathcal{R}(\mu^\star) \leq O(1)$, *meaning that Alice can lose at most* $O(1)$ *in fair zero-sum games. At the same time, our algorithm guarantees that* $\sum_{t=1}^T \mathbb{E}\left[V(\mu^t, \nu^t)\right] \leq \min_{\mu \in \Delta'} \sum_{t=1}^T \mathbb{E}\left[V(\mu, \nu^t)\right] + \tilde{O}(\sqrt{T})$, *where* $\Delta'$ *is the sim-*

plex restricted to $\mathcal{A}'$. *This means that if Bob plays suboptimally, Alice can still guarantee to win* $\Theta(T)$ *whenever these actions allow her to do so (while* $\mu^\star$ *itself does not guarantee this), as we indeed observe in Section 5.*

**Our Algorithm.** In this section, we present Algorithm 1 and explain its key steps. Our algorithm is inspired by the *Phased Aggression* algorithm, originally proposed by Even-Dar et al. (2008) for the *full-information setting*. We briefly note that a direct application of existing full-information algorithms is not possible. This is because, in the bandit setting, Alice only observes her realized cost and not the cost of the other possible actions she could have chosen. We will thus combine the phasing idea of Even-Dar et al. (2008) with appropriately importance-weighted estimators of the full cost function. Note that the same adaptation would not yield our result for full-information algorithms other than Phased Aggression.

We now give an outline of Algorithm 1. In every round $t$, the Phased Aggression algorithm plays a convex combination between the comparator strategy $\mu^c$ and the strategy $\hat{\mu}^t$ chosen by a no-regret algorithm (which runs in parallel). That is, the played strategy is $\mu^t = \alpha\hat{\mu}^t + (1 - \alpha)\mu^c$ for some $\alpha \in (0, 1]$. Whenever the algorithm estimates that the comparator $\mu^c$ is a poor choice, it increases $\alpha$ by a factor of two (so that it puts less weight on $\mu^c$ and more on the no-regret iterates) and restarts the no-regret algorithm. We group all rounds according to these restarts and call them *phases* $k = 1, 2, \ldots$. During each phase, $\alpha$ is constant.

Within this phasing scheme, the specifics of our algorithm are as follows. The no-regret algorithm of our choice is on-line mirror descent (Hazan, 2019, OMD) with the standard KL divergence $D_{\mathrm{KL}}(\mu||\mu') := \sum_a \mu(a)\log(\mu(a)/\mu'(a))$ as regularizer. In every round $t$, the algorithm plays its current action $a^t \sim \mu^t$ and observes its cost (Line 4). It uses this to construct an importance-weighted estimator $\hat{c}^t$ of the (unobserved) full cost function (Line 5). The algorithm then performs one iteration of OMD with the estimated costs (Line 11). This procedure is repeated until a new phase is started (Line 6), which happens if the comparator $\mu^c$ is performing poorly under the estimated $\hat{c}^t$'s of the current phase.

Regarding computation, the OMD update can be implemented in closed form as $\hat{\mu}^{t+1}(a) \propto \hat{\mu}^t(a)\exp(-\eta'\hat{c}^t(a))$. We can check the if-condition in Line 6 by directly computing the maximum in $O(A)$ time.

**Regret Analysis.** In this section we provide a proof sketch of Theorem 1. We defer the full proof to Appendix B.1.

We first introduce some notation. We index the variables by their respective phase $k \geq 1$: Phase $k$ lasts from $\text{start}_k$ to $\text{start}_{k+1} - 1$ and uses linear combinations with $\alpha^k = \min\{1, 2^{k-1}/R\}$ (Lines 7, 9). By design, there are at most

---

**Algorithm 1** Phased Aggression with Importance-Weighting

---

**Require:** Number of rounds $T$, comparator margin $\delta$, regret upper bound $R \leftarrow \delta^{-1}\sqrt{2T\log(A)}$, OMD learning rates $\eta \leftarrow \sqrt{\delta^2 \log(A)/(2T)}$, $\tau \leftarrow \sqrt{2\log(A)/(AT)}$.

1: Initialize $\hat{\mu}^1(a) = \mu^1(a) \leftarrow \frac{1}{A}$ for all $a \in [A]$, initialize $\alpha \leftarrow 1/R$, start $\leftarrow 1$, $k \leftarrow 1$ (counts phase).
2: **for** round $t = 1, \ldots, T$ **do**
3:      **Alice** chooses $\mu^t \in \Delta_A$, **Bob** selects cost $c^t$.                         ▷ in NFGs: $c^t = U e_{b^t}$
4:      **Alice** suffers and observes cost $c^t(a^t)$ for $a^t \sim \mu^t$.
5:      **Alice** builds cost estimator $\hat{c}^t(a) \leftarrow \frac{c^t(a^t)}{\mu^t(a)} \mathbb{1}\{a^t = a\}$.
6:      **if** $\max_{\mu \in \Delta_A} \sum_{j=\text{start}}^{t} \langle \hat{c}^j, \mu^c - \mu \rangle > 2R$ **and** $\alpha < 1$ **then**
7:          $k \leftarrow k+1$, start $\leftarrow t+1$.                  ▷ If comparator performs poorly, new phase
8:          $\hat{\mu}^{t+1}(a) \leftarrow \frac{1}{A}$ for all $a \in [A]$.                      ▷ Re-initialize OMD
9:          Update $\alpha \leftarrow \min\{2^{k-1}/R, 1\}$.                 ▷ Increase $\alpha$ for upcoming phase
10:     **else**                                               ▷ OMD update
11:          $\hat{\mu}^{t+1} \leftarrow \arg\min_{\mu \in \Delta_A} (\eta' \langle \mu, \hat{c}^t \rangle + D_{\text{KL}}(\mu||\hat{\mu}^t))$,   with $\eta' = \eta$ if $\alpha < 1$, and $\eta' = \tau$ if $\alpha = 1$.
12:     $\mu^{t+1} \leftarrow \alpha \hat{\mu}^{t+1} + (1-\alpha)\mu^c$.                 ▷ Play shifted OMD to $\mu^c$ by $1-\alpha$

---

$1 + \lceil \log_2(R) \rceil$ phases, where $R$ is a known regret upper bound for OMD input to the algorithm. The overall regret is at most the sum of regrets across all phases, and we will thus analyze each phase separately. To this end, let

$$\hat{\mathcal{R}}^k(\mu) := \sum_{t=\text{start}_k}^{\text{start}_{k+1}-1} \langle \hat{c}^t, \mu^t - \mu \rangle$$

denote the *estimated regret* during phase $k$. By convention, start$_{k+1} := T+1$ if $k$ is the last phase. The following lemma bounds this estimated regret for phases with $\alpha^k < 1$.

**Lemma 3.1** (During normal phases). *Let $k$ be such that $\alpha^k < 1$. Then for all $\mu \in \Delta_A$,*

$$\hat{\mathcal{R}}^k(\mu) \leq 2R + 2 = 2\delta^{-1}\sqrt{2T\log(A)} + 2,$$

*and for the special comparator $\hat{\mathcal{R}}^k(\mu^c) \leq 2^{k-1}$.*

The first part of the theorem establishes a worst-case bound on the estimated regret. Such a bound would normally not be possible for importance-weighted cost estimators. In our case, during phases with $\alpha^k < 1$, we put constant weight on the comparator strategy $\mu^c$, which in turn is lower bounded by $\delta > 0$. Our estimated costs (Line 5) will thus be upper bounded, which is a key step in the proof. The second part of the theorem easily follows using the definition of $\alpha^k$.

Next, suppose the algorithm exits a phase $k$ as the if-condition in Line 6 holds. The following lemma establishes that exiting the phase is justified in the sense that we perform sufficiently well compared to the special comparator, according to the estimated costs.

**Lemma 3.2** (Exiting a phase). *Let $k$ be such that $\alpha^k < 1$. If Algorithm 1 exits phase $k$, then $\hat{\mathcal{R}}^k(\mu^c) \leq -2^{k-1}$.*

We are now ready to prove Theorem 3.1. First, consider the case that $\alpha = 1$ is never reached. Note that our cost estimates are unbiased, i.e. $\mathbb{E}[\hat{c}^t(a)] = c^t(a)$. It is thus sufficient if we can bound $\hat{\mathcal{R}}^k$. As there are $O(\log R)$ phases, Lemma 3.1 implies $\max_\mu \mathcal{R}(\mu) \leq O(R \log R)$. Moreover, the previous two lemmas geometrically balance the regret compared to $\mu^c$ to be at most 1, and we conclude. Second, suppose now that $\alpha = 1$ is reached. The final phase will then simply be OMD with standard importance-weighting (a.k.a. Exp3), as we put no weight on the special comparator $\mu^c$. While we cannot apply Lemma 3.1, we can directly bound the remaining *expected* regret of Exp3 (Orabona, 2019). We can thus use the same argument as before, with one additional phase.

### 3.3. Lower Bound

We will now show that regarding the guarantee we provided in Theorem 3.1, a multiplicative dependence on the inverse of the "exploration gap" $\delta$ is indeed unavoidable.

**Theorem 3.2.** *Let $\delta \in (0, 1/A)$. There is a comparator $\mu^c \in \Delta_A$ with all $\mu^c(a) \geq \delta$ such that for any algorithm for Protocol 2 there is a sequence $c^1, \ldots, c^T \in [0,1]^A$ such that: If $\mathcal{R}(\mu^c) \leq O(1)$, then*

$$\max_{\mu \in \Delta_A} \mathcal{R}(\mu) \geq \Omega(\sqrt{\delta^{-1}T} - \delta^{-3/4}T^{1/4}).$$

The key idea of our proof is that any algorithm with low regret compared to $\mu^c = (1-\delta, \delta)$ for $A = 2$ actions will need to play action 1 most of the time if one can information-theoretically not detect that action 2 is, in fact, minimally better. We defer the proof to Appendix B.2. Finally, we remark that if the cost functions are stochastic rather than adversarial, we can match this lower bound up to logarithmic factors, see Appendix B.3.

# 4. Extensive-Form Games

In this section, we present our results for EFGs. We start by giving the definition of EFGs, using the notation that appeared in Kozuno et al. (2021); Bai et al. (2022); Fiegel et al. (2023a;b), see Appendix C.1 for a brief discussion. For clarity, we present the *two-player zero-sum* case, although our results readily generalize to arbitrary EFGs.

**Definition 4.1.** *A two-player zero-sum EFG is a tuple* $(H, \mathcal{S}, \mathcal{X}, \mathcal{Y}, \mathcal{A}, \mathcal{B}, P, x, y, u)$, *where*

- *there are* $2$ *players, Alice and Bob.* $\mathcal{A} = [A]$ *and* $\mathcal{B} = [B]$ *denote their respective sets of possible actions.*
- $\mathcal{S}$ *denotes the set of states of the game.* $H \in \mathbb{N}$ *is the horizon of the game. At stage* $h \in [H]$, $\mathcal{S}_h \subseteq \mathcal{S}$ *denotes the possible states.*
- $P := (p_0, p)$ *is the transition kernel; the game's state is sampled according to* $s_{h+1} \sim p(\cdot|s_h, a_h, b_h)$ *upon actions* $(a_h, b_h) \in \mathcal{A} \times \mathcal{B}$ *in state* $s_h \in \mathcal{S}$. *The initial state is sampled according to* $s_1 \sim p_0 \in \Delta_{\mathcal{S}}$.
- $u(s, a, b) \in [-1, 1]$ *is Alice's random* cost *(and Bob's* reward*) for actions* $(a, b) \in \mathcal{A} \times \mathcal{B}$ *chosen in state* $s \in \mathcal{S}$, *with mean* $\bar{u}(s, a, b)$.
- *Alice observes information sets (infosets) from* $\mathcal{X}$ *(*$|\mathcal{X}| = X$*), and Bob from* $\mathcal{Y}$. *Alice's infosets are described by a surjective function* $x \colon \mathcal{S} \to \mathcal{X}$ *(resp.* $y \colon \mathcal{S} \to \mathcal{Y}$ *for Bob).*

The idea behind infosets is that Alice has imperfect information about the state of the game: she cannot differentiate between sates $s, s' \in \mathcal{S}$ that belong at the same infoset, i.e. when $x(s) = x(s')$. The same holds for Bob with $y$ in lieu of $x$. This is reflected in the definition of the policy sets.

**Definition 4.2.** *A policy is a mapping* $\pi \colon \mathcal{X} \to \Delta_A$. *We denote the set of all such policies by* $\Pi$. *The policy set* $\Pi'$ *for Bob consists of all mappings* $\pi' \colon \mathcal{Y} \to \Delta_B$.

We let $\pi(a|x)$ denote the probability of playing action $a \in \mathcal{A}$ in states $s \in \mathcal{S}$ from infoset $x = x(s) \in \mathcal{X}$. As Alice cannot differentiate between states $s, s' \in \mathcal{S}$ from the same infoset, she must act the same way if $x(s) = x(s')$. Similarly, for Bob we write $\pi'(b|y)$ for $y \in \mathcal{Y}$.

**Definition 4.3.** *Given policies* $(\pi_A, \pi_B) \in \Pi \times \Pi'$, *the* expected total cost *for Alice equals*

$$V(\pi_A, \pi_B) := \mathbb{E}\left[\sum_{h=1}^{H} u(s_h, a_h, b_h)\right],$$

*where* $(s_h, a_h, b_h)$ *are the state and actions at stage* $h \in [H]$ *via* $a_h \sim \pi_A(\cdot|x(s_h))$, $b_h \sim \pi_B(\cdot|y(s_h))$, *and* $s_{h+1} \sim p(\cdot|s_h, a_h, b_h)$.

For the remainder of the section, we make the following assumptions, which are standard in the EFG literature (Kozuno et al., 2021; Bai et al., 2022; Fiegel et al., 2023a;b).

**Assumption 4.1.** • *Tree structure: For any state* $s_h \in \mathcal{S}_h$, *there exists a unique sequence* $(s_1, a_1, b_1, \ldots, s_{h-1}, a_{h-1}, b_{h-1})$ *leading to* $s_h$.
- *Perfect recall: Let* $s, s'$ *be such that* $x(s) = x(s')$. *Then:*
  – *There exists* $h \in [H]$ *such that* $s, s' \in \mathcal{S}_h$.
  – *Let* $(s_1, a_1, \ldots, s_{h-1}, a_{h-1})$ *be the unique path leading to* $s$ *and* $(s'_1, a'_1, \ldots, s'_{h-1}, a'_{h-1})$ *the unique path leading to* $s'$. *Then for all* $k \in [h-1]$: $x(s_k) = x(s'_k)$ *and* $a_k = a'_k$.

*The analogous assumption holds for* $y$ *in lieu of* $x$.

*Tree structure* states that the game proceeds in rounds during which the players cannot loop back to a previous state. We remark that this also justifies not explicitly indexing the transitions, rewards, policies, and treeplex strategies by steps $h$ to cover non-stationary dynamics. *Perfect recall* establishes that the players never forget the history of play. They can only consider two states as the same infoset if the observations so far have been the same (Hoda et al., 2010). The latter implies that infosets are partitioned along the horizon, i.e. $\mathcal{X} = \biguplus_{h \in [H]} \mathcal{X}_h$, and the same holds for $\mathcal{Y}$ and the states.

**Online Learning in EFGs.** Now suppose Alice and Bob repeatedly play an EFG for $T$ consecutive rounds. In each round $t \in [T]$, Alice and Bob select a pair of policies $(\pi_A^t, \pi_B^t) \in \Pi \times \Pi'$. Then a trajectory $(s_1^t, a_1^t, b_1^t, u_1^t, \ldots, s_H^t, a_H^t, b_H^t, u_H^t)$ is sampled according to the policies $(\pi_A^t, \pi_B^t)$ and Alice suffers cost $\sum_{h=1}^{H} u_h^t$, as summarized in Protocol 3.

---

**Protocol 3** Bandit Feedback over Policies (EFGs)

**Require:** A comparator policy $\pi^c \in \Pi$.
    **for** round $t = 1, \ldots, T$ **do**
        **Alice** selects $\pi_A^t \in \Pi$, **Bob** selects $\pi_B^t \in \Pi'$.
        **Alice** obtains costs $\sum_{h=1}^{H} u_h^t$ and observes trajectory $(x_1^t, a_1^t, u_1^t, \ldots, x_H^t, a_H^t, u_H^t)$.

---

We remark that in EFGs, we are naturally in the *bandit feedback* setting as Alice only observes the trajectory $(x_1^t, a_1^t, u_1^t, \ldots, x_H^t, a_H^t, u_H^t)$. Under *full-information feedback*, Alice would observe Bob's actual policy $\pi_B^t \in \Pi'$.

**Remark 4.1** (Importance of bandit feedback in EFGs)**.** *In EFGs, bandit feedback is considerably more natural than full-information feedback. This is due to the fact that when playing against Bob, the realized samples are only observed along* one single trajectory *in the game tree. Observing full information would thus mean knowing Bob's counterfactual policy in states that have never been visited during play, which is not realistic.*

## 4.1. From EFGs to Online Linear Minimization

As mentioned, we once more resort to the more general OLM problem. Yet this time, our strategy polytope will be the so-called treeplex $\mathcal{P} = \mathcal{T}$ rather than the simplex. The following definition provides an equivalent characterization of a policy. It will allow us to view the expected cost $V(\pi_A^t, \pi_B^t)$ as a (bi-)linear function (Hoda et al., 2010).

**Definition 4.4.** *A vector $\mu \in \mathbb{R}^{X \cdot A}$ belongs to the treeplex $\mathcal{T}$ iff for all $x_h \in \mathcal{X}_h$ and $a \in \mathcal{A}$,*

$$\begin{cases} \mu(x_h, a) \geq 0, \\ \sum_{a_h \in \mathcal{A}} \mu(x_h, a_h) = \mu(x_{h-1}, a_{h-1}), \end{cases} \quad (1)$$

*where $(x_{h-1}, a_{h-1})$ is the unique predecessor pair reaching $x_h$. We consider $\mu(x_0, a_0) = 1$ for the root by convention. We define the treeplex $\mathcal{T}'$ over Bob's infosets $\mathcal{Y}$ analogously.*

**Remark 4.2.** *There is the following equivalence between Definitions 4.2 and 4.4. Given a policy $\pi \in \Pi$, we can define a unique $\mu_\pi \in \mathcal{T}$ by $\mu_\pi(x_h, a_h) = \prod_{h'=1}^h \pi(a_{h'}|x_{h'})$, where the $(x_{h'}, a_{h'})$ form the unique path to $(x_h, a_h)$. Vice-versa, given $\mu \in \mathcal{T}$, we can recover the corresponding policy via $\pi_\mu(a|x) = \mu(x, a) / \sum_{a'} \mu(x, a')$. The same equivalence holds between Bob's policies $\Pi'$ and treeplex strategies $\mathcal{T}'$.*

By convention, we thus identify policies $(\pi_A^t, \pi_B^t)$ with their corresponding treeplex strategies $(\mu^t, \nu^t)$ and write $V(\mu^t, \nu^t)$ for Alice's expected cost. The following lemma (Kozuno et al., 2021) shows that this definition indeed allows us to view Protocol 3 as a (safe) OLM problem (Protocol 1).

**Lemma 4.1.** *For any state $s \in \mathcal{S}_h$, infoset $x = x(s) \in \mathcal{X}_h$ and action $a \in \mathcal{A}$, let $(s_1, a_1, b_1, \ldots, s_{h-1}, a_{h-1}, b_{h-1})$ be the unique path leading to $s$. Let $p(s) := p_0(s_1) \prod_{1 \leq h' \leq h-1} p(s_{h'+1}|s_{h'}, a_{h'}, b_{h'})$, and consider*

$$c^t(x, a) := \sum_{\substack{s:\, x(s)=x, \\ b \in \mathcal{B}}} p(s) \cdot \nu^t(y(s), b) \cdot \bar{u}(s, a, b). \quad (2)$$

*Then $V(\mu, \nu^t) = \langle \mu, c^t \rangle$ for all $\mu \in \mathcal{T}$.*

Alice does not observe the full cost function $c^t$, as we are in the bandit feedback setting. Yet, this lemma establishes that Protocol 1 over the treeplex $\mathcal{P} = \mathcal{T}$ covers EFGs. Thus, it is sufficient to solve the safe OLM problem (OLM).

## 4.2. Upper Bound

As in the simplex case, our Algorithm 2 guarantees Equation (OLM) for any policy $\mu^c \in \mathcal{T}$ that is $\delta$-bounded away from the boundary of the strategy polytope. Once more, we can resort to a restricted action set to relax this assumption (Remark 3.1). The result itself applies to any EFG with tree structure and perfect recall and is not restricted to the zero-sum or two-player case, since we can simply modify the costs in Equation (2) accordingly.

**Theorem 4.1.** *Let $\delta \in (0, 1/A]$. For any special comparator $\mu^c \in \mathcal{T}$ such that $\mu^c(x, a) \geq \delta$ for all $x$, $a$, Algorithm 2 achieves (for any $c^t$'s from Equation (2))*

$$\mathcal{R}(\mu^c) \leq 1, \quad \text{and} \quad \max_{\mu \in \mathcal{T}} \mathcal{R}(\mu) \leq \tilde{O}\left(\delta^{-1} \sqrt{X H^3 T}\right).$$

If the EFG is a fair zero-sum game, Alice can now choose a min-max equilibrium $\mu^c = \mu^\star$ as the comparator. If $\mu^\star$ has full support, the reduction from Section 2 then shows that Alice achieves the best of both worlds guarantee from Question 1.

**Remark 4.3.** *The dependence on $X$ is as good as desired in the sense that there is a $\sqrt{XAT}$ lower bound in the unconstrained case. The dependence on $H$ is less crucial for many relevant EFGs, as we often have $X \simeq A^H$ and so $H$ is a logarithmic factor. See Bai et al. (2022).*

**Our Algorithm.** Algorithm 2 is similar to our algorithm for the simplex. It combines the Phased Aggression scheme with importance-weighted OMD. However, in the EFG case, we have to generalize these notions to the treeplex.

In particular, we use OMD with the so-called *dilated* KL divergence as regularizer (Line 11). As we will see in the regret analysis, to this end it is crucial that we use an *unbalanced* dilated KL divergence $D$ (Kozuno et al., 2021) in the phases with $\alpha < 1$ and a *balanced* KL divergence $D^{\text{bal}}$ (Bai et al., 2022) if $\alpha = 1$ is reached. In Appendix C.2, we formally define the divergences and confirm that they allow for an efficient closed-form implementation. This is crucial as we want to avoid costly projections onto the treeplex by any means. Moreover, we can efficiently check Line 6 via standard dynamic programming over the set of policies (or solving an LP over the treeplex).

**Regret Analysis.** Our analysis follows a similar argument as in Section 3 and we defer the proofs to Appendix C.3. The main technical challenge is to transfer the regret bounds for importance-weighted OMD from the simplex (with KL) to the treeplex $\mathcal{T}$ (with dilated KL).

In addition, we now require a careful analysis to obtain a mild dependence on the number of infosets $X$ and actions $A$, in the following sense. First, when upper bounding the estimated regret in analogy to Lemma 3.1 ($\alpha < 1$), we analyze OMD with the unbalanced dilated KL divergence by adapting the argument of Kozuno et al. (2021) to our importance-weighting. Using the (more sophisticated) balanced KL here would introduce an additional undesired factor of $\sqrt{A}$. Second, once $\alpha = 1$ in the final phase, we analyze the expected regret of *balanced* OMD instead, by adapting the argument of Bai et al. (2022) to our cost estimators. Using the unbalanced divergence would introduce an extra factor of $\sqrt{X}$, which can be prohibitively large.

---

**Algorithm 2** Phased Aggression with Importance-Weighting for EFGs

---

**Require:** Number of rounds $T$, comparator margin $\delta$, regret bound $R \leftarrow \delta^{-1}\sqrt{8XH^3\log(A)T}$, learning rate $\eta \leftarrow \sqrt{\delta^2 X\log(A)/(8H^2T)}$, balanced learning rate $\tau \leftarrow \sqrt{XA\log(A)/(H^3T)}$.

1: Initialize $\hat{\mu}^1(x_h, a) = \mu^1(x_h, a) \leftarrow \frac{1}{A^h}$ ($h \in [H]$, $x_h \in \mathcal{X}_h$, $a \in \mathcal{A}$), $\alpha \leftarrow 1/R$, start $\leftarrow 1$, $k \leftarrow 1$ (counts phase).
2: **for** round $t = 1, \dots, T$ **do**
3:     **Alice** chooses $\mu^t \in \mathcal{T}$, **Bob** selects strategy $\nu^t \in \mathcal{T}'$.     ▷ and thus cost $c^t$ via Equation (2)
4:     **Alice** obtains costs $\sum_{h=1}^H u_h^t$, observes trajectory $(x_1^t, a_1^t, u_1^t, \dots, x_H^t, a_H^t, u_H^t)$.     ▷ $V(\mu^t, \nu^t)$ in expectation
5:     **Alice** builds cost estimator $\hat{c}^t(x_h, a) \leftarrow \frac{\mathbb{1}\{(x_h^t, a_h^t)=(x_h, a)\}u_h^t}{\mu^t(x_h, a)}$.
6:     **if** $\max_{\mu \in \mathcal{T}} \sum_{j=\text{start}}^t \langle \hat{c}^j, \mu^c - \mu \rangle > 2R$ **and** $\alpha < 1$ **then**
7:         start $\leftarrow t+1$, $k \leftarrow k+1$.     ▷ If comparator performs poorly, next phase
8:         $\hat{\mu}^{t+1}(x_h, a) \leftarrow \frac{1}{A^h}$ ($h \in [H]$, $x_h \in \mathcal{X}_h$, $a \in \mathcal{A}$).     ▷ Initialize to uniform policy
9:         Update $\alpha \leftarrow \min\{2^{k-1}/R, 1\}$.     ▷ Increase $\alpha$ for upcoming phase
10:     **else**     ▷ OMD update
11:

$$\hat{\mu}^{t+1} \leftarrow \begin{cases} \arg\min_{\mu \in \mathcal{T}} \left(\eta \langle \mu, \hat{c}^t \rangle + D(\mu || \hat{\mu}^t)\right) & (\text{if } \alpha < 1), \\ \arg\min_{\mu \in \mathcal{T}} \left(\tau \langle \mu, \hat{c}^t \rangle + D^{\text{bal}}(\mu || \hat{\mu}^t)\right) & (\text{if } \alpha = 1). \end{cases} \tag{3}$$

12:     $\mu^{t+1} \leftarrow \alpha\hat{\mu}^{t+1} + (1-\alpha)\mu^c.$     ▷ Play shifted OMD to $\mu^c$ by $1-\alpha$

---

### 4.3. Lower Bound

As in the case of NFGs, we show that our guarantees for Algorithm 2 are close to being tight for EFGs of arbitrary depth. Our proof reduces an EFG of depth $H$ to the simplex case from Theorem 3.2. See Appendix C.5 for the proof.

**Theorem 4.2.** *Let $A \geq 2$, $H \geq 1$, and $\delta \in (0,1)$. There exists an EFG of depth $H$ with $X = \Theta(A^H)$ such that for any $\mu^c \in \mathcal{T}$ with $\min_{x,a} \mu^c(x, a) = \delta$, there is an adversary such that for any algorithm: If $\mathcal{R}(\mu^c) \leq O(1)$, then*

$$\max_{\mu \in \mathcal{T}} \mathcal{R}(\mu) \geq \Omega(\sqrt{\delta^{-1}T} - \delta^{-3/4}T^{1/4}).$$

## 5. Experimental Evaluations

We experimentally compare our Algorithm 2 for EFGs to the standard OMD algorithm with dilated KL (Kozuno et al., 2021) as well as to minimax play. Our evaluations confirm our theoretical findings, revealing that Algorithm 2 can achieve the best of both no-regret algorithms and minimax play. They also show that our motivating question from Section 1 is indeed of practical relevance. We provide further details and evaluations in Appendix D.

**Kuhn Poker.** We consider *Kuhn poker* (Kuhn, 1950), which serves as a simple yet fundamental example of two-player zero-sum imperfect information EFGs. Kuhn poker is a common 3-card simplification of standard poker, where each player selects one card from the deck {Jack, Queen, King} without replacement and initially bets one unit.[3]

---

[3] https://en.wikipedia.org/wiki/Kuhn_poker

**Remark 5.1.** *The min-max equilibrium of Kuhn Poker is not full-support ($\delta = 0$ in Theorem 4.1). As seen in Remark 3.1, we can easily circumvent this issue by considering only the actions in the support of the equilibrium. For Kuhn Poker, this results in $\delta = 1/3$. Algorithm 2 is then still guaranteed not to lose anything while being able to compete with the best response within the support of the equilibrium.*

We consider the following baseline algorithms Alice could play over $T$ rounds of Kuhn poker:

1) play the *Min-Max* equilibrium $\pi^\star$ in every round; or
2) run *OMD* with dilated KL; or
3) run *Algorithm 2* with comparator policy $\pi^\star$.

We consider two types of experiments: First, we run the three algorithms against each other to check which of the algorithms risks losing units to others (*All vs All*). Second, we evaluate how well each algorithm allows Alice to exploit exploitable strategies (*All vs Exploitable Strategies*). We repeat each experiment 5 times.

**All vs All.** In Figure 2 we plot the total amount (of units) each algorithm *wins*. As Figure 2 shows, both Min-Max and Algorithm 2 never incur losses while both gain a significant amount against OMD. Indeed, as (symmetrized) Kuhn poker is a symmetric zero-sum game, the min-max equilibrium is guaranteed not to lose. The same holds for our Algorithm 2. In contrast, a no-regret algorithm such as OMD can lose up to $O(\sqrt{T})$ units. Interestingly, it does lose a similar amount against our Algorithm 2.

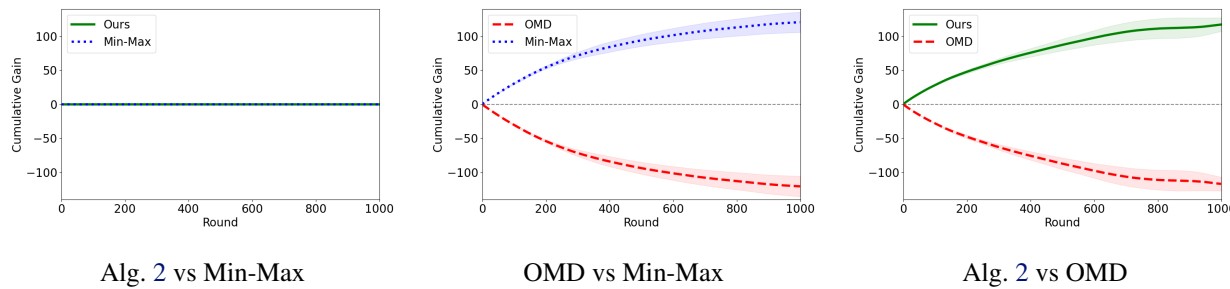

Alg. 2 vs Min-Max          OMD vs Min-Max          Alg. 2 vs OMD

Figure 2: All vs all comparison for $T = 1000$ rounds. The x-axis displays the round $t$, and the y-axis displays how much the respective algorithm (Min-Max, OMD, Algorithm 2) *gained* from the other.

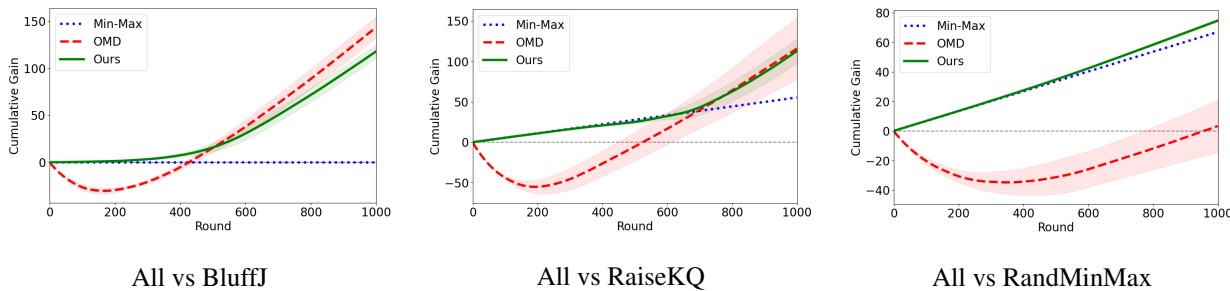

All vs BluffJ          All vs RaiseKQ          All vs RandMinMax

Figure 3: All vs Bob comparison for $T = 1000$ rounds. The x-axis displays the round $t$, and the y-axis displays how much Min-Max, OMD, and Algorithm 2 *gained* from the second algorithm so far. The y-axes have varying scales for readability.

**All vs Exploitable Strategies.** We now compare the performance of Min-Max, OMD and Algorithm 2 against the following reasonable but suboptimal strategies. The goal is to understand their ability to exploit weak strategies. We consider:

a) *BluffJ*: Bob plays the min-max equilibrium, except that he bets (bluffs) when he has a Jack;

b) *RaiseKQ*: Bob raises/calls if and only if he has a King or a Queen, and checks/folds otherwise;

c) *RandMinMax*: Each round, with probability $0.2$, he plays the uniform strategy, and else the min-max one.

In Figure 3 we present the amount (of units) each algorithm *wins* against these exploitable strategies. We first consider *All vs BluffJ & All vs RaiseKQ*. Algorithm 2 plays conservatively and gains an amount similar to Min-Max until it takes off and starts exploiting Bob near-optimally, as OMD would. OMD, in turn, first loses a certain amount of money and only matches the gain of Min-Max after exploring sufficiently, then having the same slope as Algorithm 2. The min-max equilibrium itself does not exploit *BluffJ* at all and exploits *RaiseKQ* sub-optimally. In these cases, our algorithm suffers neither of the two drawbacks of losing money or not exploiting the weak strategy. Finally, in *All vs RandMinMax*, our Algorithm 2 improves slightly over Min-Max. OMD gains at the same rate after losing an initial amount to its opponent.

## 6. Conclusion

In this paper, we showed how to provably exploit suboptimal strategies with essentially no expected risk in repeated zero-sum games by combining regret minimization and minimax play. More generally, we believe that our novel results for adversarial bandits leading to these guarantees may be of independent interest. We hope that our work inspires future research on safe online learning, including settings like convex-concave games, learning with feedback graphs, and establishing no-swap-regret guarantees.

## Impact Statement

This paper presents work whose goal is to advance the field of Machine Learning. There are many potential societal consequences of our work, none of which we feel must be specifically highlighted here.

## Acknowledgments

Adrian Müller would like to thank Gergely Neu for previous discussions regarding the stochastic case. Stratis Skoulakis was supported by the Villum Young Investigator Award (Grant no. 72091). This work is funded (in part) through a PhD fellowship of the Swiss Data Science Center, a joint venture between EPFL and ETH Zurich. This work was supported by Hasler Foundation Program: Hasler Responsible AI (project number 21043). Research was sponsored by the

Army Research Office and was accomplished under Grant Number W911NF-24-1-0048. This work was supported by the Swiss National Science Foundation (SNSF) under grant number 200021_205011.

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

# A. Further Related Work

**Safe Opponent Exploitation.** While there have been some approaches to safe learning in games (Ponsen et al., 2011; Farina et al., 2019; Zhang & Sandholm, 2021; Bernasconi-de Luca et al., 2021; Bernasconi et al., 2022; Ge et al., 2024), all these works are fairly different from our learning problem. Related to Ganzfried & Sandholm (2015); Ganzfried & Sun (2018), the works of Damer & Gini (2017); Liu et al. (2022) provide algorithms that interpolate between being safe and exploitive through a specific parameter. However, these algorithms may incur up to $\Omega(T)$ regret compared to the best fixed strategy in hindsight. Recently, Maiti et al. (2023) proved the first *instance-dependent* poly-logarithmic regret bound for noisy $2 \times 2$ NFGs, which naturally relates to our desired regret bound. However, such bounds become vacuous when the game matrix does not have pairwise distinct entries and assume to observe the opponent's action (which corresponds to full information in our feedback model).

**Exploiting Adaptive Opponents.** If Bob is oblivious and plays a fixed sequence of (mixed) strategies, then any regret Alice incurs is potential utility she could gain by playing a no-regret strategy (e.g., the best-of-both-worlds strategy we present). However, if Bob is adaptive, switching to a no-regret strategy does not necessarily allow Alice to recover additional utility (Bob could, for example, react to this by playing his minimax strategy). There is a line of recent work (Deng et al., 2019; Mansour et al., 2022; Kolumbus & Nisan, 2022b;a; Brown et al., 2023; Cai et al., 2023; Chen & Lin, 2023; Haghtalab et al., 2024; Ananthakrishnan et al., 2024; Guruganesh et al., 2024; Arunachaleswaran et al., 2024) on how to play against sub-optimal adaptive strategies (e.g. other learning algorithms) in various settings, although almost all of this work only pertains to general sum games. It is an interesting open question to understand to what extent we can obtain similar best-of-both-worlds results for adaptive opponents in zero-sum games.

**Comparator-Adaptive OL with Full Information.** In online learning (OL) under full information feedback, Hutter et al. (2005); Even-Dar et al. (2008); Kapralov & Panigrahy (2011); Koolen (2013); Sani et al. (2014) establish (with various emphases) that safe OLM over the simplex in the sense of Equation (OLM) is possible. Using so-called parameter-free methods from the online convex optimization literature instead (Orabona & Pál, 2016; Cutkosky & Orabona, 2018; Orabona, 2019, e.g.), one can (after a simple shifting argument) achieve similar guarantees in the full information setting. For our purposes, the most notable of the above algorithms is the Phased Aggression template of Even-Dar et al. (2008), as it is the only one we were able to adapt to the bandit feedback setting while maintaining the rate-optimal regret guarantee. While the application of the above type of algorithms to fair zero-sum (normal-form) games is direct (Section 2), we are not aware of any prior work making this connection, even under full-information feedback.

**Comparator-Adaptive OL with Bandit Feedback.** Lattimore (2015) establishes a sharp separation between full information and bandit feedback. The author shows that $O(1)$ regret compared to a single comparator action implies a worst-case regret of $\Omega(AT)$ for some other action. This rules out algorithms that resolve our question even in the simple normal-form case under bandit feedback. The key to this lower bound is that the algorithm has to play the special comparator essentially every time, thereby not exploring any other options (as the comparator strategy is deterministic) and thus not knowing whether it is safe to switch the arm. The minimal assumption we can make on the comparator strategy is thus that it plays every action with a non-zero probability. In addition to the mentioned works from the online convex optimization literature, van der Hoeven et al. (2020) remarkably analyzes bandit convex optimization algorithms that adapt to the comparator. However, unlike in the full information case, it is not possible to turn them into an algorithm for safe OLM (as the shifting argument one can use for full-information parameter-free methods like Orabona & Pál (2016); Cutkosky & Orabona (2018); Orabona (2019) does no longer work under bandit feedback).

**Relation to Safe Reinforcement Learning.** A closely related line of work is that of *conservative bandits* (Wu et al., 2016) and *conservative RL* (Garcelon et al., 2020). In conservative exploration, algorithms are designed to obtain at least a $(1-\alpha)$-fraction of the return of a comparator, which in our motivating example, however, means that the algorithm may suffer a linear loss $\alpha T$ in the worst case. We thus believe that independently of our motivation from a game-theoretic viewpoint, our results nicely complement existing OL literature. In constrained (or safe) reinforcement learning (Badanidiyuru et al., 2018; Efroni et al., 2020), both the regret and the cumulative violation of a constraint are considered. However, even in the stochastic case the goal of constant regret compared to some known strategy can only be realized if there exists a strategy with a strictly larger return (Liu et al., 2021) for the environment, and in the adversarial case even this reduction fails.

**OL in (Extensive-Form) Games.** While online learning (OL) in NFGs can readily be reduced to the problem of learning from experts (Cesa-Bianchi & Lugosi, 2006) (full information) or multi-armed bandits (Lattimore & Szepesvári, 2020), it becomes more difficult in the case of EFGs (Osborne & Rubinstein, 1994) due to the presence of (imperfectly observed) states and transitions. State-of-the-art algorithms for no-regret learning in EFGs are based on online mirror descent (OMD)

over the treeplex, which leads to near-optimal regret bounds in the full information setting (Farina et al., 2021; Fan et al., 2024) and the bandit setting (Farina et al., 2021; Kozuno et al., 2021; Bai et al., 2022; Fiegel et al., 2023a). Alternative approaches are based on counterfactual regret minimization (Zinkevich et al., 2007; Lanctot et al., 2009), which however do not guarantee a bound on the actual regret (see Bai et al. (2022, Theorem 7)).

## B. Deferred Proofs for Normal-Form Games

### B.1. Upper Bound

First, note that our cost estimates are unbiased, i.e. $\mathbb{E}\left[\hat{c}^t(a)\right] = \mathbb{E}\left[c^t(a)\right]$, and $\mathbb{E}\left[\langle \hat{c}^t, \mu^t\rangle\right] = \mathbb{E}\left[\mathbb{E}\left[\langle \hat{c}^t, \mu^t\rangle \mid \mathcal{F}_{t-1}\right]\right] = \mathbb{E}\left[\langle c^t, \mu^t\rangle\right] = \mathbb{E}\left[c^t(a^t)\right]$, where $\mathcal{F}_{t-1}$ is the $\sigma$-algebra induced by all random variables prior to sampling $a^t$. Further, WLOG we assume that the cost functions are bounded via $c^t \in [0,1]^A$. The reduction from NFGs with matrix entries $U_{a,b} \in [-1,1]$ is then simply via $c^t(a) := (1 + U_{a,b^t})/2$, where the shifting and scaling does not change the regret bound. By convention $\text{start}_{k+1} := T + 1$ if $k$ is the last phase.

**Theorem 3.1.** *Let $\delta \in (0, 1/A)$. Consider any mixed strategy $\mu^c \in \Delta_A$ such that $\mu^c(a) \geq \delta$ for all $a \in [A]$. Under bandit feedback (Protocol 2), for any sequence of $c^t \in [0,1]^A$, Algorithm 1 achieves*

$$\mathcal{R}(\mu^c) \leq 1, \quad \text{and} \quad \max_{\mu \in \Delta_A} \mathcal{R}(\mu) \leq \tilde{O}\left(\delta^{-1}\sqrt{T}\right).$$

*Proof. Case 1: $\alpha = 1$ is not reached.* Suppose first the algorithm ends in phase $k < 1 + \log_2(R)$ at time step $T$. By Lemma 3.1, w.r.t. any comparator

$$\sum_{t=1}^{T}\left\langle \hat{c}^t, \mu^t - \mu \right\rangle \leq (2R + 2) \cdot k \leq O(R\log(R)).$$

All previous phases must have been exited, so by Lemmas 3.1 and 3.2 we have

$$\sum_{t=1}^{T}\left\langle \hat{c}^t, \mu^t - \mu^c \right\rangle \leq 2^{k-1} - \sum_{i=1}^{k-1} 2^{i-1} = 2^{k-1} - (2^{k-1} - 1) = 1.$$

Taking expectation yields the claim.

*Case 2: $\alpha = 1$ is reached.* Next, suppose the phase $\alpha^k = 1$ was reached and simply Exp3 was run in the final phase $k$. As before

$$\sum_{t=1}^{\text{start}_k - 1}\left\langle \hat{c}^t, \mu^t - \mu \right\rangle \leq (2R + 2) \cdot k \leq O(R\log(R)).$$

For the final phase, note that Algorithm 1 plays Exp3 for $\leq T$ rounds, with uniform initialization. By the standard Exp3 analysis (Orabona, 2019, Sec. 10.1), this phase has expected regret

$$\mathbb{E}\left[\sum_{t=\text{start}_{k+1}}^{T}\left\langle c^t, \mu^t - \mu \right\rangle\right] \leq \frac{\log(A)}{\tau} + \frac{\tau}{2}AT \leq \sqrt{AT\log(A)/2} \leq \delta^{-1}\sqrt{2\log(A)T} = R. \tag{4}$$

since $\tau = \sqrt{\frac{2\log(A)}{AT}}$ and $\delta \leq 1/A$. Thus for any comparator $\mu \in \Delta_A$ we have

$$\mathbb{E}\left[\sum_{t=1}^{T}\left\langle c^t, \mu^t - \mu \right\rangle\right] \leq O(R\log(R)).$$

Finally, for the special comparator note that all phases $k' < k$ have been left and thus by Lemma 3.2 and Equation (4)

$$\mathbb{E}\left[\sum_{t=1}^{T}\left\langle c^t, \mu^t - \mu^c \right\rangle\right] \leq R - \sum_{k'=1}^{k-1} 2^{k'-1} = R - (2^{k-1} - 1) \leq 1,$$

where the last step used that $\alpha^k = \min\{1, 2^{k-1}/R\} = 1$ and thus $R \leq 2^{k-1}$. □

Recall that

$$\hat{\mathcal{R}}^k(\mu) := \sum_{t=\text{start}_k}^{\text{start}_{k+1}-1} \langle \hat{c}^t, \mu^t - \mu \rangle = \alpha^k \sum_{j=\text{start}_k}^{\text{start}_{k+1}-1} \langle \hat{c}^t, \hat{\mu}^t - \mu \rangle + (1-\alpha^k) \sum_{t=\text{start}_k}^{\text{start}_{k+1}-1} \langle \hat{c}^t, \mu^c - \mu \rangle \tag{5}$$

measures Alice's estimated regret.

**Lemma 3.1** (During normal phases). *Let $k$ be such that $\alpha^k < 1$. Then for all $\mu \in \Delta_A$,*

$$\hat{\mathcal{R}}^k(\mu) \leq 2R + 2 = 2\delta^{-1}\sqrt{2T\log(A)} + 2,$$

*and for the special comparator $\hat{\mathcal{R}}^k(\mu^c) \leq 2^{k-1}$.*

*Proof.* WLOG suppose that $R = 2^r$ is a power of 2, else we can run the algorithm for $T$ such that $R$ is the next largest power of two and pay a constant factor in the regret. For the first term in Equation (5), we analyze OMD to bound $\sum_{t=\text{start}_k}^{\text{start}_{k+1}-1} \langle \hat{c}^t, \hat{\mu}^t - \mu \rangle$ almost surely, making use of the fact that $\hat{c}^t$ is bounded. Indeed, recall

$$\hat{c}^t(a) = \frac{c^t(a^t)}{\mu^t(a)} \mathbb{1}\left\{a^t = a\right\} \leq \frac{1}{\mu^t(a)}.$$

We have $\alpha^k = 2^{k-1}/R \leq 2^{\log_2(R)-1}/R = 1/2$, so

$$\hat{c}^t(a) \leq \frac{1}{\mu^t(a)} = \frac{1}{\alpha^k \mu^t(a) + (1-\alpha^k)\mu^c(a)} \leq \frac{1}{\frac{1}{2}\mu^c(a)} \leq \frac{2}{\delta}.$$

Moreover, $\hat{c}^t$ is zero outside the visited $a^t$. Thus, by Lemma B.1, almost surely for the first term in Equation (5)

$$\sum_{t=\text{start}_k}^{\text{start}_{k+1}-1} \langle \hat{c}^t, \hat{\mu}^t - \mu \rangle \leq \frac{\log(A)}{\eta} + 2\eta T\delta^{-2} \leq \delta^{-1}\sqrt{2T\log(A)} = R. \tag{6}$$

For the second term in Equation (5), note that since the if condition may only hold at $t' := \text{start}_{k+1} - 1$,

$$\sum_{t=\text{start}_k}^{\text{start}_{k+1}-1} \langle \hat{c}^t, \mu^c - \mu \rangle \leq 2R + \frac{c^{t'}(a^{t'})}{\frac{1}{2}\mu^c(a^{t'})}\mu^c(a^{t'}) \leq 2R + 2. \tag{7}$$

Linearly combining Equations (6) and (7),

$$\hat{\mathcal{R}}^k(\mu) := \alpha^k \sum_{t=\text{start}_k}^{\text{start}_{k+1}-1} \langle \hat{c}^t, \hat{\mu}^t - \mu \rangle + (1-\alpha^k) \sum_{t=\text{start}_k}^{\text{start}_{k+1}-1} \langle \hat{c}^t, \mu^c - \mu \rangle \leq 2R + 2$$

for any $\mu$. For the special comparator, by Equation (6)

$$R^k(\mu^c) = \alpha^k \sum_{t=\text{start}_k}^{\text{start}_{k+1}-1} \langle \hat{c}^t, \hat{\mu}^t - \mu^c \rangle + (1-\alpha^k) \sum_{t=\text{start}_k}^{\text{start}_{k+1}-1} \langle \hat{c}^t, \mu^c - \mu^c \rangle \leq (2^{k-1}/R)R = 2^{k-1}.$$

$\square$

**Lemma B.1** (OMD with bounded surrogate costs). *Let $\eta > 0$, and $L > 0$. Let $(\hat{c}^t)_t$ be cost functions such that for all $t$, $0 \leq \hat{c}^t(a) \leq L$ (for all $a$), and moreover $\hat{c}^t(a) = 0$ if $a \neq a^t$ for some arbitrary $a^t$. Set $\hat{\mu}^1(a) = 1/A$ and consider the scheme $\mu^{t+1} = \arg\min_{\mu \in \Delta_A} \langle \mu, \hat{c}^t \rangle + \frac{1}{\eta}D(\mu||\hat{\mu}^t)$ for $t \leq T'$. Then we have for all $\mu \in \Delta_A$*

$$\sum_{t=1}^{T'} \langle \hat{\mu}^t - \mu, \hat{c}^t \rangle \leq \frac{\log(A)}{\eta} + \frac{\eta}{2}L^2T'.$$

*Proof.* From Orabona (2019, Sec. 10.1), we find that a.s.

$$\sum_{t=1}^{T'} \left\langle \hat{\mu}^t - \mu, \hat{c}^t \right\rangle \le \frac{\log(A)}{\eta} + \frac{\eta}{2} \sum_{t=1}^{T'} \sum_a \hat{\mu}^t(a)(\hat{c}^t(a))^2 \le \frac{\log(A)}{\eta} + \frac{\eta}{2} \sum_{t=1}^{T'} \hat{\mu}^t(a^t) L^2 \le \frac{\log(A)}{\eta} + \frac{\eta}{2} L^2 T'.$$

$\square$

**Lemma 3.2** (Exiting a phase). *Let $k$ be such that $\alpha^k < 1$. If Algorithm 1 exits phase $k$, then $\hat{\mathcal{R}}^k(\mu^c) \le -2^{k-1}$.*

*Proof.* At $t = \text{start}_{k+1} - 1$ the if condition implies $\max_{\mu \in \Delta_A} \sum_{j=\text{start}_k}^{\text{start}_{k+1}-1} \left\langle \hat{c}^j, \mu^c - \mu \right\rangle > 2R$, so when we let $\mu^\star$ be a maximizer, we find

$$\sum_{t=\text{start}_k}^{\text{start}_{k+1}-1} \left\langle \hat{c}^t, \mu^t - \mu^c \right\rangle = \alpha^k \sum_{t=\text{start}_k}^{\text{start}_{k+1}-1} \left\langle \hat{c}^t, \hat{\mu}^t - \mu^c \right\rangle$$

$$= \alpha^k \sum_{t=\text{start}_k}^{\text{start}_{k+1}-1} \left\langle \hat{c}^t, \hat{\mu}^t - \mu^\star \right\rangle + \alpha^k \sum_{t=\text{start}_k}^{\text{start}_{k+1}-1} \left\langle \hat{c}^t, \mu^\star - \mu^c \right\rangle$$

$$\le \alpha^k R + \alpha^k(-2R)$$

$$= -2^{k-1},$$

where we used Equation (6) in the last inequality. $\square$

## B.2. Lower Bound

**Theorem 3.2.** *Let $\delta \in (0, 1/A]$. There is a comparator $\mu^c \in \Delta_A$ with all $\mu^c(a) \ge \delta$ such that for any algorithm for Protocol 2 there is a sequence $c^1, \ldots, c^T \in [0,1]^A$ such that: If $\mathcal{R}(\mu^c) \le O(1)$, then*

$$\max_{\mu \in \Delta_A} \mathcal{R}(\mu) \ge \Omega(\sqrt{\delta^{-1}T} - \delta^{-3/4}T^{1/4}).$$

Our lower bound becomes vacuous in the regime where $\delta \le O(T^{-1})$, which is when a direct application of Lattimore (2015) shows a trivial $\Omega(T)$ lower bound.

*Proof.* It is sufficient to prove the lower bound for $A = 2$ actions as we can assign the same distribution to all but one action. We prove a lower bound for stochastic cost functions, which immediately implies the same bound for adversarially chosen costs. Consider the following setup with two different environments. The first action deterministically gives cost $c_1 = 1/2$ in both environments. In the *first environment* $(-)$, action two samples costs according to a $\text{Ber}(\frac{1}{2} - \gamma T^{-1/2})$ distribution with expected cost $c_- = \frac{1}{2} - \gamma T^{-1/2}$. We will choose $\gamma > 0$ later and for now, only require $\gamma < \frac{1}{2}T^{1/2}$ in order for the sampling to be well-defined. Symmetrically, in the *second environment* $(+)$, action two samples costs according to a $\text{Ber}(\frac{1}{2} + \gamma T^{-1/2})$ distribution with expected cost $c_+ = \frac{1}{2} + \gamma T^{-1/2}$. We consider the case that the *special comparator* is $\mu^c = (1 - \delta, \delta) \in \Delta_2$. In the following, $\mathcal{R}(\mu)$ denotes the regret compared to $\mu \in \Delta_A$ in the worst case environment. We fix an arbitrary algorithm and index regret and expectation with $+$ or $-$ to indicate which probability space (environment) we are referring to.

Now let $N_2$ be the number of times action two is chosen during the $T$ interactions. The requirement on the regret w.r.t. the special comparator (together with the standard regret decomposition (Lattimore & Szepesvári, 2020, Sec. 4.5)) shows

$$1 \ge \mathcal{R}(\mu^c) \ge \mathcal{R}_+(\mu^c) = \mathbb{E}_+ [N_2] (+\gamma T^{-1/2}) - \delta T(+\gamma T^{-1/2}),$$

and thus

$$\mathbb{E}_+ [N_2] \le \gamma^{-1} T^{1/2} + \delta T.$$

Plugging this into Lemma B.2, we have (if $\gamma < \frac{1}{2\sqrt{2}}T^{1/2}$)

$$
\begin{aligned}
\mathbb{E}_-[N_2] \leq & \mathbb{E}_+[N_2] + T\sqrt{2}\sqrt{\mathbb{E}_+[N_2]}\gamma T^{-1/2} \\
\leq & (\gamma^{-1}T^{1/2} + \delta T) + T\sqrt{2}\sqrt{\gamma^{-1}T^{1/2} + \delta T}\gamma T^{-1/2} \\
\leq & \gamma^{-1}T^{1/2} + \delta T + T\sqrt{2}(\gamma^{-1/2}T^{1/4} + \delta^{1/2}T^{1/2})\gamma T^{-1/2} \\
\leq & \gamma^{-1}T^{1/2} + \delta T + \sqrt{2}\gamma^{1/2}T^{3/4} + \sqrt{2}\delta^{1/2}\gamma T
\end{aligned}
$$

Using this in the regret decomposition on $(-)$, we see for the second action $\mu = e_2 = (0,1) \in \Delta_A$

$$
\begin{aligned}
\mathcal{R}(e_2) \geq & \mathcal{R}_-(e_2) \\
\geq & (\gamma^{-1}T^{1/2} + \delta T + \sqrt{2}\gamma^{1/2}T^{3/4} + \sqrt{2}\delta^{1/2}\gamma T)(-\gamma T^{-1/2}) - T(-\gamma T^{-1/2}) \\
\geq & -1 - \delta\gamma T^{1/2} - \sqrt{2}\gamma^{3/2}T^{1/4} - \sqrt{2}\delta^{1/2}\gamma^2 T^{1/2} + \gamma T^{1/2} \\
= & \left((1-\delta)\gamma - \sqrt{2}\delta^{1/2}\gamma^2\right)T^{1/2} - \sqrt{2}\gamma^{3/2}T^{1/4} - 1.
\end{aligned}
$$

We can now choose $\gamma = c\delta^{-1/2}$ for a sufficiently small absolute constant $c$ to show that

$$
\mathcal{R}(e_2) \geq \Theta\left(\delta^{-1/2}T^{1/2}\right) - \Theta(\delta^{-3/4}T^{1/4}). \tag{8}
$$

This bound holds when $\gamma < \frac{1}{2\sqrt{2}}T^{1/2}$, i.e. $\delta \geq c'T^{-1}$ for some large enough absolute constant $c'$. $\qquad\square$

**Lemma B.2** (Entropy inequality Bernoulli). *In the setup of Theorem 3.2, we have*

$$
\mathbb{E}_-[N_2] \leq \mathbb{E}_+[N_2] + T\sqrt{\frac{2(\gamma T^{-1/2})^2}{\frac{1}{4} - (\gamma T^{-1/2})^2}\mathbb{E}_+[N_2]}.
$$

*In particular for $\gamma < \frac{1}{2\sqrt{2}}T^{1/2}$, we have $\mathbb{E}_-[N_2] \leq \mathbb{E}_+[N_2] + \sqrt{2\mathbb{E}_+[N_2]}\gamma T^{-1/2}$.*

*Proof.* Via Pinsker's and the chain rule for the KL divergence (c.f. Auer et al. (1995) and Lattimore (2015, Appendix))

$$
\mathbb{E}_-[N_2] - \mathbb{E}_+[N_2] \leq T\sqrt{\frac{1}{2}\mathbb{E}_+[N_2] \cdot \mathrm{KL}(X\|Y)},
$$

where $X \sim \mathrm{Ber}(\frac{1}{2} + \epsilon)$ and $Y \sim \mathrm{Ber}(\frac{1}{2} - \epsilon)$ for $\epsilon = \gamma T^{-1/2}$. We conclude by computing

$$
\begin{aligned}
\mathrm{KL}(X\|Y) = & \left(\frac{1}{2} + \epsilon\right)\log\left(\frac{\frac{1}{2} + \epsilon}{\frac{1}{2} - \epsilon}\right) + \left(\frac{1}{2} - \epsilon\right)\log\left(\frac{\frac{1}{2} - \epsilon}{\frac{1}{2} + \epsilon}\right) \\
\leq & \left(\frac{1}{2} + \epsilon\right)\left(\frac{\frac{1}{2} + \epsilon}{\frac{1}{2} - \epsilon} - 1\right) + \left(\frac{1}{2} - \epsilon\right)\left(\frac{\frac{1}{2} - \epsilon}{\frac{1}{2} + \epsilon} - 1\right) \\
= & 2\epsilon\left(-\frac{\frac{1}{2} - \epsilon}{\frac{1}{2} + \epsilon} + \frac{\frac{1}{2} + \epsilon}{\frac{1}{2} - \epsilon}\right) \\
= & 2\epsilon\frac{2\epsilon}{\frac{1}{4} - \epsilon^2}.
\end{aligned}
$$

$\qquad\square$

## B.3. The Stochastic Case

As claimed in the main part, we now sketch how the $\tilde{O}(\sqrt{\delta^{-1}T})$ lower bound from Section 3.3 can be matched (up to logarithmic terms) if the costs are stochastic and not adversarial. This improves slightly over our result for the adversarial case.

**Theorem B.1.** *Let $\delta \in (0, 1)$ and consider Protocol 2 but where all $c^t(a) \sim q_a$ are i.i.d. for some fixed distributions $q_a$ with support in $[0, 1]$. Then there is an algorithm such that for any specified $\mu^c \in \Delta_A$ with all $\mu^c(a) \geq \delta$ and all distributions $q$, we have*

$$\mathcal{R}(\mu^c) \leq 1 \quad \text{and} \quad \max_{\mu \in \Delta_A} \mathcal{R}(\mu) \leq O\left(\sqrt{\delta^{-1} T \log(AT)} + \delta^{-2} \log(T)\right).$$

For $a \in [A]$, let the $a$-th action's reward distribution $q_a$ have mean $1 - m_a$, and write $m$ for the corresponding vector. We thus consider *maximization* of the *rewards* $1 - c^t$ that have means $m$. This is just for convenience to better highlight the relation of our algorithm to the classic UCB algorithm (Lattimore & Szepesvári, 2020). As for the rewards, we index the entries of a strategy $\mu \in \Delta_A$ as $\mu_a = \mu(a)$. Fix an arbitrary $a^\star \in \arg\max_a m_a$. The (random) pseudo-regret of the algorithm is

$$\tilde{\mathcal{R}} := \sum_{t=1}^{T} (m_{a^\star} - \langle \mu^t, m \rangle).$$

**Algorithm.** Construct $\underline{m}^t = (\underline{m}_1^t, \ldots, \underline{m}_A^t)$, $\overline{m}^t = (\overline{m}_1^t, \ldots, \overline{m}_A^t)$ to be the vectors of lower and upper confidence bounds for the actions after playing and observing $t$ rounds. Formally,

$$\underline{m}_a^t := \hat{m}_a^t - b_a^t, \qquad \overline{m}_a^t := \hat{m}_a^t + b_a^t,$$

where $\hat{m}_a^t$ is the average reward among the rounds in which the $a$-th action is chosen during rounds $1, \ldots, t$ (and zero if not defined), and $b_a^t$ is a confidence half-width to be specified. With this, set $M^t := [\underline{m}^t, \overline{m}^t] := [\underline{m}_1^t, \overline{m}_1^t] \times \cdots \times [\underline{m}_A^t, \overline{m}_A^t]$. Consider the following update. Let

$$\mu^1 = \mu^c,$$

and in round $t + 1$, update

$$\mu^{t+1} = \arg\max_{\mu \in \Delta_A} \min_{\tilde{m} \in [\underline{m}^t, \overline{m}^t]} \langle \mu - \mu^t, \tilde{m} \rangle. \tag{9}$$

**Regret analysis.** First, note that conditioned on $m \in M^t$, we have

$$0 = \min_{\tilde{m} \in M^t} \langle \mu^t - \mu^t, \tilde{m} \rangle \leq \max_{\mu \in \Delta_A} \min_{\tilde{m} \in M^t} \langle \mu - \mu^t, \tilde{m} \rangle = \min_{\tilde{m} \in M^t} \langle \mu^{t+1} - \mu^t, \tilde{m} \rangle \leq \langle \mu^{t+1} - \mu^t, m \rangle. \tag{10}$$

Hence, the algorithm monotonically improves, i.e. $\langle \mu^{t+1}, m \rangle \geq \langle \mu^t, m \rangle$, if all confidence intervals include the true mean. As for the confidence intervals, set $b_a^t := 2\sqrt{\frac{2 \log(T^2 A/\zeta)}{n_a^t}}$, where $n_a^t$ is the number of times that action $a$ is chosen in rounds $1, \ldots, t$. Then by Hoeffding's inequality, with probability at least $1 - \zeta$, for all $t \in [T]$ we have $m \in \text{int}(M^t)$. We call this event $G$.

By finding the closed form of the update rule in Equation (9) and the lower bound on $\mu^1 = \mu^c$, it is not hard to see the following.

**Lemma B.3.** *Conditioned on $G$, we have $\mu_{a^\star}^t \geq \mu_{a^\star}^c \geq \delta$ for all $t \in [T]$.*

Using Hoeffding's inequality and a union bound, we thus get the following concentration.

**Lemma B.4.** *Condition on $G$ and let $\zeta' \in (0, 1)$. Then with probability at least $1 - \zeta'$, we have*

$$n_{a^\star}^t \geq \delta t - \sqrt{2t \log(T/\zeta')}.$$

We are now ready to prove Theorem B.1. Condition on $G$ and on the event in Lemma B.4. This occurs with probability at least $1 - \zeta - \zeta'$.

First, we consider the regret compared to $\mu^c$. By the monotonicity property in Equation (10),

$$\langle \mu^t, m \rangle \geq \langle \mu^{t-1}, m \rangle \geq \cdots \geq \langle \mu^1, m \rangle = \langle \mu^c, m \rangle.$$

Setting $\zeta = \zeta' = \frac{1}{2T}$ and integrating out the regret of at most $T$ under the failure event:

$$\mathbb{E}\left[\sum_{t=1}^{T} \langle \mu^c - \mu^t, m \rangle\right] \leq \Pr[G]\mathbb{E}\left[\sum_t \langle \mu^c - \mu^t, m \rangle \mid G\right] + \Pr[\bar{G}]T \leq 0 + (\zeta + \zeta')T = 1.$$

We now consider the worst case (pseudo-) regret $\tilde{\mathcal{R}}$. Note that for the minimax problem in Equation (9), strong duality holds and we can fix a saddle point $(\mu^t, \tilde{m}^t)$ such that (for all $(\mu, \tilde{m}) \in \Delta_A \times M^t$)

$$\langle \mu - \mu^{t-1}, \tilde{m}^t \rangle \leq \langle \mu^t - \mu^{t-1}, \tilde{m}^t \rangle \leq \langle \mu^t - \mu^{t-1}, \tilde{m} \rangle. \tag{11}$$

Under the success events, we have $n_{a^\star}^t \geq \delta t - \sqrt{2t \log(T/\zeta')}$ by Lemma B.4. Now when $t \geq t_0 := 8\delta^{-2} \log(T/\zeta')$, then $n_{a^\star}^t \geq 2\delta^{-1}\sqrt{2t \log(T/\zeta')}$ and hence

$$b_{a^\star}^t \leq 2\sqrt{\frac{4\log(T^2 A/\zeta)}{\delta t}}. \tag{12}$$

We have

$$\tilde{\mathcal{R}} = 8\delta^{-2} \log(T/\zeta') + \sum_{t=t_0}^{T}(m_{a^\star} - \langle \mu^t, m \rangle) \leq 8\delta^{-2} \log(T/\zeta') + \sum_{t=t_0}^{T}(m_{a^\star} - \langle \mu^t, m \rangle),$$

where the instantaneous regret for $t \geq t_0$ is (with $\mu^\star := e_{a^\star}$)

$$
\begin{aligned}
m_{a^\star} - \langle \mu^t, m \rangle &= \langle \mu^\star - \mu^t, m \rangle \\
&= \langle \mu^\star - \mu^t, \tilde{m}^{t+1} \rangle + \langle \mu^\star - \mu^t, m - \tilde{m}^{t+1} \rangle \\
&\leq \langle \mu^{t+1} - \mu^t, m \rangle + \langle \mu^\star - \mu^t, m - \tilde{m}^{t+1} \rangle && \text{(by Equation (11) and } m \in M^{t+1}) \\
&\leq \langle \mu^{t+1} - \mu^t, m \rangle + \langle \mu^\star, b^{t+1} \rangle + \langle \mu^t, b^{t+1} \rangle \\
&\leq \langle \mu^{t+1} - \mu^t, m \rangle + \langle \mu^\star, b^t \rangle + \langle \mu^t, b^t \rangle. && \text{(as } b^{t+1} \leq b^t)
\end{aligned}
$$

Hence,

$$
\begin{aligned}
\tilde{\mathcal{R}} &\leq 8\delta^{-2} \log(T/\zeta') + \langle \mu^{T+1} - \mu^{t_0}, m \rangle + \sum_{t=t_0}^{T} \left( \langle \mu^\star, b^t \rangle + \langle \mu^t, b^t \rangle \right) \\
&\leq 8\delta^{-2} \log(T/\zeta') + 1 + \sum_{t=t_0}^{T} b_{a^\star}^t + \sum_{t=t_0}^{T} \langle \mu^t, b^t \rangle \\
&\leq 8\delta^{-2} \log(T/\zeta') + 1 + \sum_{t=t_0}^{T} 2\sqrt{\frac{4\log(T^2 A/\zeta)}{\delta t}} + \sum_{t=t_0}^{T} \langle \mu^t, b^t \rangle && \text{(by Equation (12))} \\
&\leq O\left( \delta^{-2} \log(T/\zeta') + \sqrt{\frac{T \log(AT/\zeta)}{\delta}} \right) + \sum_{t=t_0}^{T} \langle \mu^t, b^t \rangle,
\end{aligned}
$$

with probability at least $1 - \zeta - \zeta'$. Using $\sum_{t=t_0}^{T} \mathbb{E}[\langle \mu^t, b^t \rangle] \leq O(\sqrt{AT \log(AT/\zeta)})$ (Auer et al., 2008), $\delta \leq 1/A$ and integrating out the regret under the failure event yields the result.

## C. Deferred Proofs for Extensive-Form Games
### C.1. EFG Background
The following remark clarifies that the Markov game we defined in Section 4 (which is more common in the machine learning literature) indeed covers the case of imperfect information EFGs (which are more common in the game theory literature).

**Remark C.1.** *The notion of EFG in Definition 4.1 from Section 4 is usually referred to as a tree-structured perfect-recall partially-observable Markov game (TP-POMG). This also covers the notion of perfect-recall imperfect information extensive-form games (P-IIEFG) (Osborne & Rubinstein, 1994) that satisfy the timeability condition Jakobsen et al. (2016). In fact, a more careful look reveals that the results directly generalize to any P-IIEFG without timeability (see Bai et al. (2022) for this brief discussion).*

For further clarification, we remark that usually, both the cost function $u$, the transition probabilities $p$ and the policies $\pi$ (and treeplex strategies $\mu$) may be non-stationary in the sense that they explicitly vary across the stages $h \in [H]$ of the EFG. However, as we assume tree structure and perfect recall, the state space and infoset space are partitioned along the stages anyway, which is why WLOG we omit the explicit dependence of the above functions on the stage $h$. Finally, to be precise, our algorithm assumes to know the tree structure of the game (but not necessarily the transitions), an assumption that can be removed (Fiegel et al., 2023a).

### C.2. OMD over the Treeplex

The unbalanced and balanced dilated KL divergence are defined as follows:

$$D(\mu||\mu') := \sum_{\substack{x \in \mathcal{X}, \\ a \in \mathcal{A}}} \mu(x, a) \log\left(\frac{\pi_\mu(a|x)}{\pi_{\mu'}(a|x)}\right),$$

$$D^{\text{bal}}(\mu||\mu') := \sum_{h=1}^{H} \sum_{\substack{x_h \in \mathcal{X}_h, \\ a \in \mathcal{A}}} \frac{\mu(x_h, a)}{\mu^{h,\text{bal}}(x_h, a)} \log\left(\frac{\pi_\mu(a|x_h)}{\pi_{\mu'}(a|x_h)}\right),$$

where $\pi_\mu$ is the policy corresponding to the treeplex strategy $\mu$, and $\mu^{h,\text{bal}}$ is the unique strategy corresponding to the *balanced exploration policy*

$$\pi^{h,\text{bal}}(a|x_{h'}) := \begin{cases} \frac{|\mathcal{C}_h(x_{h'}, a)|}{|\mathcal{C}_h(x_{h'})|} & (h' \in \{1, \ldots, h-1\}), \\ \frac{1}{A} & (h' \in \{h, \ldots, H\}), \end{cases}$$

with $\mathcal{C}_h(x_{h'}, a) \subset \mathcal{X}_h$ being set of infosets at step $h$ reachable from $(x_{h'}, a)$ (i.e. the unique path to such an infoset goes through $(x_{h'}, a)$), and $|\mathcal{C}_h(x_{h'})| := \bigcup_{a \in \mathcal{A}} \mathcal{C}_h(x_{h'}, a)$.

**Computation of Unbalanced OMD.** For completeness, we restate the closed-form implementation of case one in Equation (3) with the *unbalanced* dilated divergence $D$ from Kozuno et al. (2021, Appendix B). In the setup of Equation (3), let $\hat{\pi}^t \in \Pi$ be the policy corresponding to $\hat{\mu}^t$. Then we have a closed-form

$$\hat{\pi}^{t+1}(a_h|x_h^t) = \hat{\pi}^t(a_h|x_h^t) \exp\left(\mathbb{1}\left\{a_h^t = a_h\right\}\left(-\eta \hat{c}^t(x_h^t, a_h) + \log(Z_{h+1}^t)\right) - \log(Z_h^t)\right),$$

and $\hat{\pi}^{t+1}(\cdot|x_h) = \hat{\pi}^t(\cdot|x_h)$ for all other $x_h \neq x_h^t$. Here, $Z_h^t$ is

$$Z_h^t := 1 - \hat{\pi}^t(a_h^t|x_h^t) + \hat{\pi}^t(a_h^t|x_h^t) \exp\left(-\eta \hat{c}^t(x_h^t, a_h^t) + \log(Z_{h+1}^t)\right),$$

and $Z_{H+1}^t := 1$.

**Computation of Balanced OMD.** For completeness, we also restate the closed-form implementation of case two in Equation (3) with the *balanced* dilated divergence $D^{\text{bal}}$ from Bai et al. (2022, Algorithm 5). Once more, let $\hat{\pi}^t \in \Pi$ be the policy corresponding to $\hat{\mu}^t$. Then we have a closed form for the next iterate, namely

$$\hat{\pi}^{t+1}(a_h|x_h^t) = \hat{\pi}^t(a_h|x_h^t) \exp\left(\mathbb{1}\left\{a_h = a_h^t\right\}\left(-\tau \mu^{\text{bal},h}(x_h^t, a_h^t)\hat{c}^t(x_h^t, a_h^t) + \frac{\mu^{\text{bal},h}(x_h^t, a_h^t)\log(Z_{h+1}^t)}{\mu^{\text{bal},h+1}(x_{h+1}^t, a_{h+1}^t)}\right) - \log(Z_h^t)\right),$$

and in the other infosets $\hat{\pi}^{t+1}(a_h|x_h) = \hat{\pi}^t(a_h|x_h)$. Here,

$$Z_h^t := 1 - \hat{\pi}(a_h^t|x_h^t) + \hat{\pi}(a_h^t|x_h^t) \exp\left(-\tau \mu^{\text{bal},h}(x_h^t, a_h^t)\hat{c}^t(x_h^t, a_h^t) + \frac{\mu^{\text{bal},h}(x_h^t, a_h^t)\log(Z_{h+1}^t)}{\mu^{\text{bal},h+1}(x_{h+1}^t, a_{h+1}^t)}\right),$$

and $Z_{H+1}^t = 1$.

## C.3. Upper Bound

First, note that due to the importance-weighting by the rollout policies the cost estimators are unbiased (Kozuno et al., 2021): $\mathbb{E}\left[\hat{c}^t(x,a)\right] = \mathbb{E}\left[c^t(x,a)\right]$, and $\mathbb{E}\left[\langle \hat{c}^t, \mu^t \rangle\right] = \mathbb{E}\left[\mathbb{E}\left[\langle \hat{c}^t, \mu^t \rangle \mid \mathcal{F}_{t-1}\right]\right] = \mathbb{E}\left[\langle c^t, \mu^t \rangle\right]$, where $\mathcal{F}_{t-1}$ is the $\sigma$-algebra induced by all random variables prior to sampling the trajectory $(s_1^t, a_1^t, b_1^t, u_1^t \ldots, s_H^t, a_H^t, b_H^t, u_H^t)$. Further, WLOG we assume that the costs $u(s,a,b)$ used to define the cost function $c^t$ in Equation (2) are bounded in $[0,1]$. While in EFGs we assumed $u(s,a,b) \in [-1,1]$, we can simply replace them by $(1 + u(s,a,b))/2$ without changing the regret bound. With this, we can prove the desired upper bound by resorting to the estimated regret

$$\hat{\mathcal{R}}^k(\mu) := \sum_{t=\text{start}_k}^{\text{start}_{k+1}-1} \langle \hat{c}^t, \mu^t - \mu \rangle = \alpha^k \sum_{j=\text{start}_k}^{\text{start}_{k+1}-1} \langle \hat{c}^t, \hat{\mu}^t - \mu \rangle + (1 - \alpha^k) \sum_{t=\text{start}_k}^{\text{start}_{k+1}-1} \langle \hat{c}^t, \mu^c - \mu \rangle. \tag{13}$$

By convention $\text{start}_{k+1} := T + 1$ if $k$ is the last phase.

**Theorem 4.1.** *Let $\delta \in (0, 1/A]$. For any special comparator $\mu^c \in \mathcal{T}$ such that $\mu^c(x,a) \geq \delta$ for all $x$, $a$, Algorithm 2 achieves (for any $c^t$'s from Equation (2))*

$$\mathcal{R}(\mu^c) \leq 1, \quad and \quad \max_{\mu \in \mathcal{T}} \mathcal{R}(\mu) \leq \tilde{O}\left(\delta^{-1}\sqrt{XH^3T}\right).$$

*Proof. Case 1: $\alpha = 1$ is not reached.* Suppose first the algorithm ends in phase $k$ with $\alpha^k < 1$ at time step $T$. By Lemma C.1, w.r.t. any comparator

$$\sum_{t=1}^{T} \langle \hat{c}^t, \mu^t - \mu \rangle \leq (2R + 2H) \cdot k \leq O(R \log(R)).$$

All previous phases must have been exited, so by Lemmas C.1 and C.2 we have

$$\sum_{t=1}^{T} \langle \hat{c}^t, \mu^t - \mu^c \rangle \leq 2^{k-1} - \sum_{i=1}^{k-1} 2^{i-1} = 2^{k-1} - (2^{k-1} - 1) = 1.$$

Taking expectation yields the claim.

*Case 2: $\alpha = 1$ is reached.* Next, suppose $\alpha^k = 1$ was reached. Then balanced mirror descent was run in the final phase $k$. As before

$$\sum_{t=1}^{\text{start}_k -1} \langle \hat{c}^t, \mu^t - \mu \rangle \leq (2R + 2H) \cdot k \leq O(R \log(R)).$$

For the final phase, note that the algorithm runs balanced OMD with importance weights and uniform initialization for $\leq T$ rounds. Thus by Lemma C.5, this phase has expected regret

$$\mathbb{E}\left[\sum_{t=\text{start}_k}^{T} \langle c^t, \mu^t - \mu \rangle\right] \leq \tau H^3 T + \frac{1}{\tau} D^{\text{bal}}(\mu \| \mu^{\text{start}_k})$$

$$\leq \tau H^3 T + \frac{XA \log(A)}{\tau} \quad \text{(by Bai et al. (2022), Lemma C.7))}$$

$$\leq \sqrt{XAH^3 \log(A)T} \quad \text{(since } \tau = \sqrt{\frac{XA \log(A)}{2H^3 T}}\text{)}$$

$$\leq R, \tag{14}$$

using $\delta \leq 1/A$. Thus for any comparator, we have

$$\mathbb{E}\left[\sum_{t=1}^{T} \langle c^t, \mu^t - \mu \rangle\right] \leq O(R \log(R)) + R = O(R \log(R)).$$

Finally, for the special comparator, we note that all phases $k'$ with $\alpha^{k'} < 1$ have been left and thus by Lemma C.2 and Equation (14)

$$\mathbb{E}\left[\sum_{t=1}^{T} \langle c^t, \mu^t - \mu^c \rangle\right] \leq R - \sum_{k'=1}^{k-1} 2^{k'-1} = R - (2^{k-1} - 1) \leq 1,$$

where the last step used that $\alpha^k = \min\{1, 2^{k-1}/R\} = 1$ and thus $R \leq 2^{k-1}$. $\qquad\square$

The following lemma establishes the statement from Lemma 3.1, generalized to EFGs. The second part of the lemma is essentially the same. Once more, the fact that $\mu^c$ is lower bounded comes into play when upper bounding the estimated cost functions.

**Lemma C.1** (During normal phases). *Let $k$ be such that $\alpha^k < 1$. Then for all $\mu \in \mathcal{T}$, almost surely*

$$\hat{\mathcal{R}}^k(\mu) \leq 2R + 2H = 2\delta^{-1}\sqrt{8XH^3 \log(A)T} + 2H,$$

*and for the special comparator almost surely $\hat{\mathcal{R}}^k(\mu^c) \leq 2^{k-1}$.*

*Proof.* WLOG suppose that $R = 2^r$ is a power of 2, else we can run the algorithm for $T$ such that $R$ is the next largest power of two and pay a constant factor in the regret. For the first term in Equation (13), we analyze unbalanced OMD to bound $\sum_{t=\text{start}_k}^{\text{start}_{k+1}-1} \langle \hat{c}^t, \hat{\mu}^t - \mu \rangle$ almost surely, making use of the fact that $\hat{c}^t$ is bounded. Recall

$$\hat{c}^t(x_h, a) = \frac{\mathbb{1}\{(x_h^t, a_h^t) = (x_h, a)\} u_h^t}{\mu^t(x_h, a)} \leq \frac{1}{\mu^t(x_h, a)}.$$

Now since $R = 2^r$ is a power of 2, we have $\alpha^k = 2^{k-1}/R \leq 2^{\log_2(R)-1}/R = 1/2$, so

$$\hat{c}^t(x_h, a) \leq \frac{1}{\mu^t(x_h, a)} = \frac{1}{\alpha\mu^t(x_h, a) + (1-\alpha)\mu^c(x_h, a)} \leq \frac{1}{\frac{1}{2}\mu^c(x_h, a)} \leq \frac{2}{\delta}.$$

Moreover, $\hat{c}^t$ is zero outside the visited $((x_h^t, a_h^t))_h$. Thus, by Lemma C.3, for the first term in Equation (13) almost surely

$$\sum_{t=\text{start}_k}^{\text{start}_{k+1}-1} \langle \hat{c}^t, \hat{\mu}^t - \mu \rangle \leq \frac{X \log(A)}{\eta} + \frac{4\eta TH(H+1)}{\delta^2} \leq \delta^{-1}\sqrt{8XH^2 \log(A)T} \leq R. \tag{15}$$

For the second term in Equation (13), note that since the if may only hold at $t' := \text{start}_{k+1} - 1$,

$$\sum_{t=\text{start}_k}^{\text{start}_{k+1}-1} \langle \hat{c}^t, \mu^c - \mu \rangle \leq 2R + \langle \hat{c}^{t'}, \mu^c \rangle \leq 2R + 2H. \tag{16}$$

Linearly combining Equations (15) and (16),

$$\hat{\mathcal{R}}^k(\mu) = \alpha^k \sum_{t=\text{start}_k}^{\text{start}_{k+1}-1} \langle \hat{c}^t, \hat{\mu}^t - \mu \rangle + (1-\alpha^k) \sum_{t=\text{start}_k}^{\text{start}_{k+1}-1} \langle \hat{c}^t, \mu^c - \mu \rangle \leq 2R + 2H$$

for any $\mu$, and for the special comparator $\mu^c$ we have by Equation (15)

$$R^k(\mu^c) = \alpha^k \sum_{t=\text{start}_k}^{\text{start}_{k+1}-1} \langle \hat{c}^t, \hat{\mu}^t - \mu^c \rangle + (1-\alpha^k) \sum_{t=\text{start}_k}^{\text{start}_{k+1}-1} \langle \hat{c}^t, \mu^c - \mu^c \rangle \leq (2^{k-1}/R)R = 2^{k-1}.$$

$\qquad\square$

Now suppose the algorithm exits a phase $k$. The following result mimics Lemma 3.2 for the case of EFGs, and we resort to essentially the same proof.

**Lemma C.2** (Exiting a phase). *Let $k$ be such that $\alpha^k < 1$ and suppose Algorithm 2 exits phase $k$ at time step $start_{k+1} - 1$. Then almost surely $\hat{\mathcal{R}}^k(\mu^c) \leq -2^{k-1}$.*

*Proof.* The if condition implies $\max_{\mu \in \mathcal{T}} \sum_{t=start_k}^{start_{k+1}-1} \langle \hat{c}^t, \mu^c - \mu \rangle > 2R$, so when we let $\mu^\star$ be a maximizer, we find

$$
\begin{aligned}
\sum_{t=start_k}^{start_{k+1}-1} \langle \hat{c}^t, \mu^t - \mu^c \rangle =& \alpha^k \sum_{t=start_k}^{start_{k+1}-1} \langle \hat{c}^t, \hat{\mu}^t - \mu^c \rangle \\
=& \alpha^k \sum_{t=start_k}^{start_{k+1}-1} \langle \hat{c}^t, \hat{\mu}^t - \mu^\star \rangle + \alpha^k \sum_{t=start_k}^{start_{k+1}-1} \langle \hat{c}^t, \mu^\star - \mu^c \rangle \\
\leq& \alpha^k R + \alpha^k(-2R) \\
=& -2^{k-1},
\end{aligned}
$$

using Equation (15) in the last inequality. $\square$

## C.4. Auxiliary Lemmas: OMD on the EFG Tree
## Unbalanced OMD Lemmas.

**Lemma C.3** (Bandit OMD with bounded surrogate costs). *Let $\eta > 0$, and $L > 0$. Let $(\hat{c}^t)_t$ be cost functions such that for all $t$, $0 \leq \hat{c}^t(x_h, a) \leq L$ (for all $x_h$, $a$), and moreover $\hat{c}^t(x_h, a) = 0$ if $(x_h, a) \neq (x_h^t, a_h^t)$, where $x_h^t, a_h^t$ are arbitrary. Set $\hat{\mu}^1(x_h, a) = 1/A^h$ and consider the scheme*

$$
\hat{\mu}^{t+1} = \arg\min_{\mu \in \mathcal{T}} \langle \mu, \hat{c}^t \rangle + \frac{1}{\eta} D(\mu || \hat{\mu}^t)
$$

*for $t \leq T'$. Then we have for all $\hat{\mu} \in \mathcal{T}$*

$$
\sum_{t=1}^{T'} \langle \hat{\mu}^t - \hat{\mu}, \hat{c}^t \rangle \leq \frac{X \log(A)}{\eta} + \eta H(H+1)L^2 T'.
$$

*Proof.* By Lemma C.4,

$$
\begin{aligned}
D(\hat{\mu} || \hat{\mu}^t) - D(\hat{\mu} || \hat{\mu}^{t+1}) + D(\hat{\mu}^t || \hat{\mu}^{t+1}) =& -(D(\hat{\mu} || \hat{\mu}^{t+1}) - D(\hat{\mu} || \hat{\mu}^t)) + (D(\hat{\mu}^t || \hat{\mu}^{t+1}) - D(\hat{\mu}^t || \hat{\mu}^t)) \\
=& \eta \langle \hat{\mu}^t - \hat{\mu}, \hat{c}^t \rangle.
\end{aligned}
$$

Thus (using $D \geq 0$), we have a regret bound of

$$
\sum_{t=1}^{T'} \langle \hat{\mu}^t - \hat{\mu}, \hat{c}^t \rangle \leq \frac{1}{\eta} \left( D(\hat{\mu} || \hat{\mu}^1) + \sum_{t=1}^{T} D(\hat{\mu}^t || \hat{\mu}^{t+1}) \right).
$$

For the first term we easily have $D(\hat{\mu} || \hat{\mu}^1) \leq X \log(A)$ (Kozuno et al., 2021, Lemma 6). For the second term, by Lemma C.4, we have

$$
D(\hat{\mu}^t || \hat{\mu}^{t+1}) = D(\hat{\mu}^t || \hat{\mu}^{t+1}) - D(\hat{\mu}^t || \hat{\mu}^t) \leq \eta \langle \hat{\mu}^t, \hat{c}^t \rangle + \log(Z_1^t) = \eta \sum_{h=1}^{H} \hat{\mu}^t(x_h^t, a_h^t) \hat{c}^t(x_h^t, a_h^t) + \log(Z_1^t),
$$

using that $\hat{c}^t$ is zero outside $((x_h^t, a_h^t))_h$. By Equation (17) and $\log(1 + x) \leq x$,

$$
\begin{aligned}
\log(Z_1^t) \leq& \sum_{h=1}^{H} \hat{\mu}^t(x_h^t, a_h^t) \exp\left( -\eta \sum_{h'=1}^{h-1} \hat{c}^t(x_{h'}^t, a_{h'}^t) \right) \left( \exp\left( -\eta \hat{c}^t(x_h^t, a_h^t) \right) - 1 \right) \\
\leq& \sum_{h=1}^{H} \hat{\mu}^t(x_h^t, a_h^t) \exp\left( -\eta \sum_{h'=1}^{h-1} \hat{c}^t(x_{h'}^t, a_{h'}^t) \right) \left( -\eta \hat{c}^t(x_h^t, a_h^t) + \eta^2 \hat{c}^t(x_h^t, a_h^t)^2 \right),
\end{aligned}
$$

where we used $\exp(-y) \leq 1 - y + y^2$ for $y \geq 0$. We thus find, using $\hat{c}^t \geq 0$ throughout,

$$
\begin{aligned}
D(\hat{\mu}^t \| \hat{\mu}^{t+1}) \leq & \eta \sum_{h=1}^{H} \hat{\mu}^t(x_h^t, a_h^t) \hat{c}^t(x_h^t, a_h^t) + \log(Z_1^t) \\
\leq & \eta \sum_{h=1}^{H} \hat{\mu}^t(x_h^t, a_h^t) \hat{c}^t(x_h^t, a_h^t) \\
& + \sum_{h=1}^{H} \hat{\mu}^t(x_h^t, a_h^t) \exp\left(-\eta \sum_{h'=1}^{h-1} \hat{c}^t(x_{h'}^t, a_{h'}^t)\right) \left(-\eta \hat{c}^t(x_h^t, a_h^t) + \eta^2 \hat{c}^t(x_h^t, a_h^t)^2\right) \\
= & \eta \sum_{h=1}^{H} \hat{\mu}^t(x_h^t, a_h^t) \hat{c}_h^t(x_h^t, a_h^t) \left(1 - \exp\left(-\eta \sum_{h'=1}^{h-1} \hat{c}^t(x_{h'}^t, a_{h'}^t)\right)\right) \\
& + \eta^2 \sum_{h=1}^{H} \hat{\mu}^t(x_h^t, a_h^t) \exp\left(-\eta \sum_{h'=1}^{h-1} \hat{c}^t(x_{h'}^t, a_{h'}^t)\right) \hat{c}^t(x_h^t, a_h^t)^2 \\
\leq & \eta \sum_{h=1}^{H} \hat{\mu}^t(x_h^t, a_h^t) \hat{c}^t(x_h^t, a_h^t) \left(1 - \exp\left(-\eta \sum_{h'=1}^{h-1} \hat{c}^t(x_{h'}^t, a_{h'}^t)\right)\right) \\
& + \eta^2 \sum_{h=1}^{H} \hat{\mu}^t(x_h^t, a_h^t) \hat{c}^t(x_h^t, a_h^t)^2 \\
\leq & \eta \sum_{h=1}^{H} \hat{\mu}^t(x_h^t, a_h^t) \hat{c}^t(x_h^t, a_h^t) \left(\eta \sum_{h'=1}^{h-1} \hat{c}^t(x_{h'}^t, a_{h'}^t)\right) + \eta^2 \sum_{h=1}^{H} \hat{\mu}^t(x_h^t, a_h^t) \hat{c}^t(x_h^t, a_h^t)^2,
\end{aligned}
$$

where we used $1 - \exp(-x) \leq x$ in the last step. Finally, using the bound on the cost functions and the fact that all $\hat{\mu}^t(x_h^t, a_h^t) \leq 1$, we find

$$
D(\hat{\mu}^t \| \hat{\mu}^{t+1}) \leq \eta^2 H^2 L^2 + \eta^2 H L^2 \leq \eta^2 H(H+1) L^2.
$$

Summing over $t$ concludes the proof. $\qquad\square$

In the setup of Lemma C.3, let $\hat{\pi}^t \in \Pi$ be the policy corresponding to $\hat{\mu}^t$ and recall (Appendix C.1)

$$
\begin{aligned}
Z_{H+1}^t = & 1, \\
Z_h^t = & \sum_{a_h} \hat{\pi}^t(a_h | x_h^t) \exp\left(\mathbb{1}\{a_h^t = a_h\}\left(-\eta \hat{c}^t(x_h^t, a_h) + \log(Z_{h+1}^t)\right)\right) \\
= & 1 - \hat{\pi}^t(a_h^t | x_h^t) + \hat{\pi}^t(a_h^t | x_h^t) \exp\left(-\eta \hat{c}^t(x_h^t, a_h^t) + \log(Z_{h+1}^t)\right),
\end{aligned}
$$

The following lemma is a slight generalization of Kozuno et al. (2021). Indeed, the proof only uses that $\hat{c}^t$ is zero outside of the visited $((x_h^t, a_h^t))_h$, not whether we normalize by $\hat{\mu}^t$ or $\mu^t$ or from which policy the trajectory $(x_h^t, a_h^t)_h$ is sampled from. The same holds for the following closed form of $Z_1^t$ (Kozuno et al., 2021, c.f. Lemma 6):

$$
Z_1^t = 1 + \sum_{h'=1}^{H} \hat{\mu}^t(x_{h'}^t, a_{h'}^t) \exp\left(-\eta \sum_{h''=1}^{h'-1} \hat{c}^t(x_{h''}^t, a_{h''}^t)\right) \left(\exp\left(-\eta \hat{c}^t(x_{h'}^t, a_{h'}^t)\right) - 1\right). \tag{17}
$$

**Lemma C.4** (Kozuno et al. (2021), Lemma 7). *In the setup of Lemma C.3, we have*

$$
D(\mu \| \hat{\mu}^{t+1}) - D(\mu \| \hat{\mu}^t) = \eta \langle \mu, \hat{c}^t \rangle + \log(Z_1^t)
$$

*a.s. for all $t \leq T'$, $\mu \in \mathcal{T}$.*

**Balanced OMD Lemmas.** Recall the definition of $c^t$ from Equation (2), for which $\hat{c}^t$ is an unbiased estimator. Again, recall that we WLOG replaced assume $u(s, a, b) \in [0, 1]$ (by rescaling via $(1 + u(s, a, b))/2$) for simplicity, without changing the regret bound.

**Lemma C.5.** *Let $\tau > 0$. Set $\hat{\mu}^1(x_h, a) = 1/A^h$ and with costs from Equation (2) for Protocol 3 consider the scheme*

$$\hat{c}^t(x_h, a) = \frac{\mathbb{1}\left\{(x_h, a) = (x_h^t, a_h^t)\right\} u_h^t}{\hat{\mu}^t(x_h, a)},$$

$$\hat{\mu}^{t+1} = \arg\min_{\mu \in \mathcal{T}} \left(\langle \mu, \hat{c}^t \rangle + \frac{1}{\tau} D^{bal}(\mu || \hat{\mu}^t)\right)$$

*for $t \leq T'$. Then for all $\hat{\mu} \in \mathcal{T}$*

$$\mathbb{E}\left[\sum_{t=1}^{T'} \langle \hat{\mu}^t - \hat{\mu}, c^t \rangle\right] \leq \frac{\tau}{2} H^3 T' + \frac{1}{\tau} D^{bal}(\mu || \mu^1).$$

*Proof.* By Lemma C.6, we have

$$\frac{1}{\tau} \left(D^{\text{bal}}(\hat{\mu} || \hat{\mu}^{t+1}) - D^{\text{bal}}(\hat{\mu} || \hat{\mu}^t)\right) = \langle \hat{\mu}, \hat{c}^t \rangle + \Xi_1^t.$$

Thus,

$$\frac{1}{\tau} \mathbb{E}\left[D^{\text{bal}}(\hat{\mu} || \hat{\mu}^{T'}) - D^{\text{bal}}(\hat{\mu} || \hat{\mu}^1)\right] = \mathbb{E}\left[\sum_{t=1}^{T'} \langle \hat{\mu}, \hat{c}^t \rangle + \sum_{t=1}^{T'} \Xi_1^t\right]$$

$$\leq \mathbb{E}\left[\sum_{t=1}^{T'} \langle \hat{\mu} - \hat{\mu}^t, \hat{c}^t \rangle\right] + \frac{\tau H^3}{2} T' \qquad \text{(by Lemma C.7)}$$

$$= \mathbb{E}\left[\sum_{t=1}^{T'} \langle \hat{\mu} - \hat{\mu}^t, c^t \rangle\right] + \frac{\tau H^3}{2} T',$$

as $\mathbb{E}\left[\hat{c}^t(x, a) \mid \mathcal{F}_{t-1}\right] = c^t(x, a)$. Using $D^{\text{bal}} \geq 0$, we conclude

$$\mathbb{E}\left[\sum_{t=1}^{T'} \langle \hat{\mu}^t - \hat{\mu}, c^t \rangle\right] \leq \frac{1}{\tau} D^{\text{bal}}(\hat{\mu} || \hat{\mu}^1) + \frac{\tau H^3}{2} T'.$$

$\square$

As before, the following lemma from Bai et al. (2022, Lemma D.7) does not use the specific form of the cost estimates but only the update rules.

**Lemma C.6.** *In the setup of Lemma C.5, for all $\mu \in \mathcal{T}$, we have*

$$D^{bal}(\mu || \hat{\mu}^{t+1}) - D^{bal}(\mu || \hat{\mu}^t) = \tau \langle \mu, \hat{c}^t \rangle + \frac{\log(Z_1^t)}{\mu^{bal,1}(x_1^t, a_1^t)} = \tau \langle \mu, \hat{c}^t \rangle + \tau \Xi_1^t.$$

We introduce some extra notation for convenience: Let $\hat{\pi}^t \in \Pi$ be the policy corresponding to $\hat{\mu}^t$ and set

$$\beta_h^t := \tau \mu^{\text{bal},h}(x_h^t, a_h^t), \quad \hat{\pi}_h^t := \hat{\pi}^t(a_h^t | x_h^t), \quad \hat{c}_h^t := \hat{c}^t(x_h^t, a_h^t),$$

and consider the functions

$$\Xi_H^t(\hat{c}) := \Xi_H^t(\hat{c}_H) := \log\left(1 - \hat{\pi}_H^t + \hat{\pi}_H^t \exp(-\beta_H^t \hat{c}_H)\right) / \beta_H^t,$$

$$\Xi_h^t(\hat{c}) := \Xi_h^t(\hat{c}_{h:H}) := \log\left(1 - \hat{\pi}_h^t + \hat{\pi}_h^t \exp(\beta_h^t(\Xi_{h+1}^t(\hat{c}_{h+1:H}) - \hat{c}_h))\right) / \beta_H^t \quad (h < H),$$

and the values

$$\Xi_h^t := \Xi_h^t(\hat{c}^t) = \frac{1}{\beta_h^t} \log(Z_h^t) = \frac{1}{\beta_h^t} \log\left(1 - \hat{\pi}_h^t + \hat{\pi}_h^t \exp(\beta_h^t(\Xi_{h+1}^t - \hat{c}_h^t)))\right) \quad (h \in [H])$$

for the input $\hat{c}^t$. The following lemma now lets us bound the remaining term in the proof of Lemma C.5.

**Lemma C.7.** *In the setup of Lemma C.5, we have*

$$\sum_{t=1}^{T} \mathbb{E}\left[\Xi_1^t\right] \leq -\sum_{t=1}^{T'} \mathbb{E}\left[\langle \hat{\mu}^t, \hat{c}^t \rangle\right] + \frac{\tau}{2} H^3 T'.$$

*Proof.* By Lemma C.8 and as $\hat{c}^t$ is unbiased,

$$\sum_{t=1}^{T'} \mathbb{E}\left[\Xi_1^t\right] \leq -\sum_{t=1}^{T'} \mathbb{E}\left[\langle \hat{\mu}^t, \hat{c}^t \rangle\right] + \frac{\tau H}{2} \sum_{t=1}^{T'} \sum_{h=1}^{H} \sum_{h'=h}^{H} \sum_{x_{h'}, a_{h'}} \mathbb{E}\left[\mu_{1:h}^{\mathrm{bal},h}(x_h, a_h)\hat{\mu}_{h+1:h'}^t(x_{h'}, a_{h'})\hat{c}^t(x_{h'}, a_{h'})\right]$$

$$= -\sum_{t=1}^{T'} \mathbb{E}\left[\langle \hat{\mu}^t, c^t \rangle\right] + \frac{\tau H}{2} \sum_{t=1}^{T'} \sum_{h=1}^{H} \sum_{h'=h}^{H} \underbrace{\sum_{x_{h'}, a_{h'}} \mathbb{E}\left[\mu^{\mathrm{bal},h}(x_h, a_h)\hat{\mu}_{h+1:h'}^t(x_{h'}, a_{h'})c^t(x_{h'}, a_{h'})\right]}_{\leq 1}$$

$$\leq -\sum_{t=1}^{T'} \mathbb{E}\left[\langle \hat{\mu}^t, c^t \rangle\right] + \frac{\tau H^3}{2} T'.$$

$\square$

**Lemma C.8** (Bai et al. (2022), Lemma D.11). *We have*

$$\Xi_1^t \leq -\langle \hat{\mu}^t, \hat{c}^t \rangle + \frac{\tau H}{2} \sum_{h=1}^{H} \sum_{h'=h}^{H} \sum_{x_{h'}, a_{h'}} \mu^{bal,h}(x_{h'}, a_{h'})\hat{\mu}_{h+1:h'}^t(x_{h'}, a_{h'})\hat{c}^t(x_{h'}, a_{h'}),$$

*where* $\hat{\mu}_{h+1:h'}^t(x_{h'}^t, a_{h'}^t) := \prod_{h''=h+1}^{h'} \hat{\pi}^t(a_{h''}|x_{h''})$ *along the unique path* $(x_{h''}, a_{h''})_{h''}$ *leading from step* $h+1$ *to* $(x_{h'}^t, a_{h'}^t)$.

The proof is the same as in Bai et al. (2022).[4]

### C.5. Lower Bound

**Theorem 4.2.** *Let* $A \geq 2$, $H \geq 1$, *and* $\delta \in (0,1)$. *There exists an EFG of depth* $H$ *with* $X = \Theta(A^H)$ *such that for any* $\mu^c \in \mathcal{T}$ *with* $\min_{x,a} \mu^c(x,a) = \delta$, *there is an adversary such that for any algorithm: If* $\mathcal{R}(\mu^c) \leq O(1)$, *then*

$$\max_{\mu \in \mathcal{T}} \mathcal{R}(\mu) \geq \Omega(\sqrt{\delta^{-1}T} - \delta^{-3/4}T^{1/4}).$$

*Proof.* Consider an $A$-nary tree with $X = \Theta(A^H)$ leaves and where each infoset corresponds to a unique state. As for the transitions, the learner is deterministically sent to a leaf $s = (a_1, \ldots, a_A)$ upon playing $a_h$ in each step $h$. Since $\delta = \min_{x,a} \mu^c(x,a)$, there also exists a leaf information set $x = x(s)$ and and an action $a$ such that $\mu^c(x,a) = \delta$. Now consider two environments in which all state-action triples have cost one, except for the cost in leaf $s$, which is either sampling according to the $(+)$ or $(-)$ environment from Theorem 3.2. We are thus effectively simulating a two-armed bandit with comparator $(1-\delta, \delta)$ with the same construction as in the simplex case. The derivation in Theorem 3.2 thus concludes the proof. $\square$

---

[4]There, in (ii) we still have $\hat{\mu}^t(x_h, a_h)\hat{c}^t(x_h, a_h) \leq 1$. All other properties used in the proof hold for general $\hat{c} \geq 0$ (in particular Lemma D.9 and D.10, although stated for $\tilde{\ell} \in [0,1]^H$).

## D. Further Experimental Evaluations

In this section, we provide further details regarding our experimental evaluations in Section 5.

**All vs Exploitable Strategies.** In addition to Section 5, we compare the performance of Min-Max, OMD and Algorithm 2 against a couple of other exploitable strategies strategies. We consider the following constant strategies:

- *RaiseK*: Bob raises/calls if and only he has a King, and checks/folds otherwise.
- *RandMinMax($\alpha$)*: Bob plays a perturbed version of the Min-Max strategy: In every round, with a small probability $\alpha$, he will play the uniform strategy, and otherwise the Min-Max strategy.

In Figure 4, we present the amount of money that each of Min-Max, OMD, and Algorithm 2 extract with respect to the aforementioned exploitable strategies. Specifically, Figure 4 reveals the following.

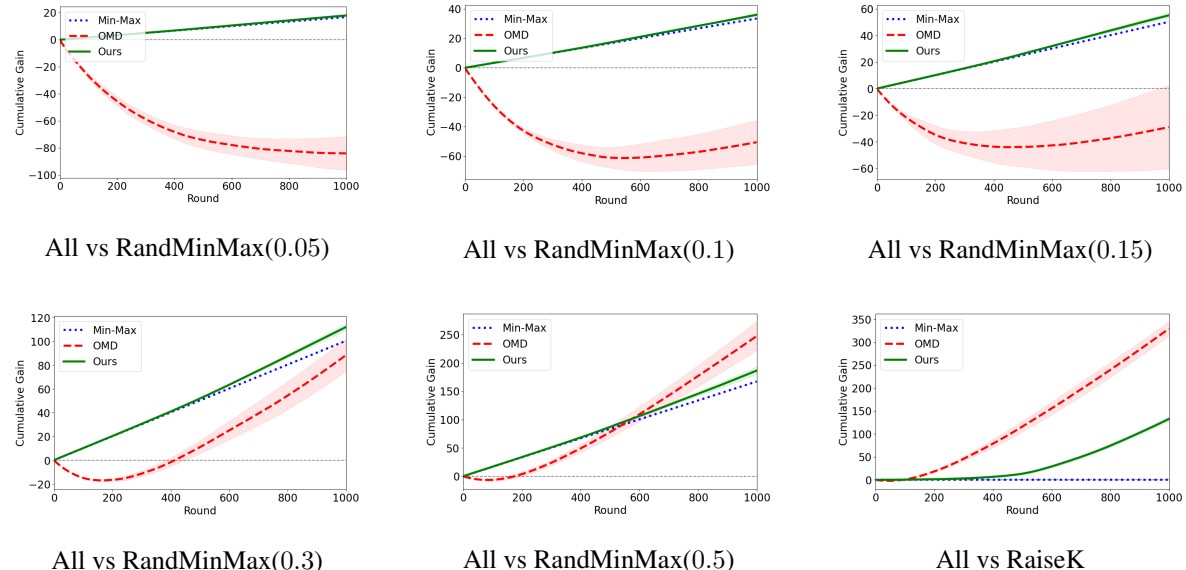

Figure 4: All vs Bob comparison for $T = 1000$ rounds. The x-axis displays the round $t$, and the y-axis displays how much Min-Max, OMD, and Algorithm 2 gained from the second algorithm so far. The y-axes have varying scales for readability.

*All vs RandMinMax($\alpha$)*: In all plots, our Algorithm 2 achieves at least the gain of the min-max equilibrium and in fact always improves slightly over it. For small values of $\alpha$ (e.g. $\alpha = 0.05$), meaning that Bob plays a (reasonable) strategy very close to the min-max equilibrium, OMD always loses money while Algorithm 2 wins linearly. For larger values of $\alpha$ (e.g. $\alpha = 0.1, 0.15, 0.3$), OMD loses an initial amount but slowly starts catching up towards a total positive gain for very large $T$. Finally, when $\alpha$ is large (e,g, $\alpha = 0.5$), meaning that Bob plays a highly suboptimal (and not exploitative) strategy, OMD is able to obtain a positive gain much quicker and eventually surpasses our Algorithm 2 (as it is not restricted to the support of the min-max equilibrium, which in this case is of advantage).

*All vs RaiseK*: Notice that min-max equilibrium does not exploit RaiseK at all. At the same time, OMD exploits it linearly right away, extracting a near-optimal gain from the opponent. Our Algorithm 2 also exploits *RaiseK* linearly at a comparable slope, however starting exploitation somewhat delayed due to the risk-averse nature of the algorithm. However, our algorithm consistently exploits weak opponents significantly better than the min-max strategy in all cases, and unlike OMD does so while not risking to lose essentially any money.

In summary, our experimental evaluations reveal the following insights that are in accordance with our theoretical findings: If Alice plays Algorithm 2, she secures at least the gain of the min-max strategy, thus not losing against any opponent. Yet, she is able to better exploit strategies that deviate from the min-max strategy, at a level often comparable to standard no-regret algorithms.

**Implementation Details.** In all experiments, we average $n = 5$ runs of repeated play (plotting Alice's average cumulative expected gain), and plot one standard deviation. In all algorithms, we used the same learning fixed rates ($\eta \sim 1/\sqrt{T}$) and the (unbalanced) dilated KL divergence for fairness and simplicity. We provide the code in the supplementary material.

