# OpenReview forum: "Best of Both Worlds: Regret Minimization versus Minimax Play"
_ICML.cc/2025/Conference — ICML 2025 poster_

### Official Review · Reviewer_aZNc · 2025-02-17

**Overall Recommendation:** 4

**Summary:**

This submission studies whether an algorithm can achieve O(1) regret compared to a specific fixed comparitor strategy while also guaranteeing O(T^0.5) regret compared to the best strategy in hindsight in symmetric two-player zero-sum games. The authors focus on bandit feedback settings, where the learner only observes the cost of their chosen action. The authors answer this answer affirmatively in symmetric two-player zero-sum games by introducing algorithms that nagivate the trade-off between no-regret learning and minimax equilibrium strategies in both normal-form and extensive-form games.

The authors’ algorithm is a generalization of the “phased aggression” approach of Even-Dar et al. 2008, which was originally proposed for the full information setting. At a high level, the algorithm plays a convex combination between the comparator strategy and the strategy chosen by a no-regret algorithm at each time-step. Whenever the algorithm estimates that the comparator is a poor choice, it decreases the weight on the comparator strategy (thereby increasing the weight on the no-regret strategy). The authors use importance weighting to estimate the losses of each action, given the bandit feedback present. The no-regret algorithm used is online mirror descent with the KL divergence regularizer.

The regret guarantees for this algorithm suffer a multiplicative dependence on a term corresponding to the “exploration gap”. However, the authors prove a lower bound which shows that this dependence is indeed necessary. The authors also extend their results to handle extensive-form games. Their algorithm for this setting is similar, but uses online mirror descent with an unbalanced dilated KL divergence regualrizer as the no-regret algorithm. As was the case in normal-form games, the authors show that their performance guarantees are “nearly tight” by providing a corresponding lower bound.

Finally, the authors empirically evaluate their algorithm for extensive-form games on Kuhn poker and compare its performance to standard no-regret learning and minimax play. Their empirical evaluations largely confirm their theoretical findings, i.e. the proposed results gives “best of both worlds” performance.

**Claims And Evidence:**

Yes.

**Essential References Not Discussed:**

No.

**Experimental Designs Or Analyses:**

Yes.

**Methods And Evaluation Criteria:**

Yes.

**Other Comments Or Suggestions:**

n/a

**Other Strengths And Weaknesses:**

The authors are the first to provide compelling results for best of both worlds learning in two-player zero-sum games under bandit feedback. Their algorithms are natural generalizations of those for the full information setting, and they provide nearly matching upper- and lower-bounds. A minor weakness is the restriction to zero-sum games that are symmetric, although there are many zero-sum games which satisfy this property.

**Questions For Authors:**

What are the barriers preventing you from extending your results to non-symmetric zero-sum games? Or to no-swap-regret?

**Relation To Broader Scientific Literature:**

This submission belongs to the literature on learning and equilibrium computation in games. Specifically, this work is the first to study best-of-both-worlds performance under bandit feedback in symmetric, zero-sum games.

**Theoretical Claims:**

Yes.

---

> ### Author Rebuttal · Authors · 2025-03-30
>
> Thank you for carefully reviewing our paper and providing valuable feedback!
>
> ## **Remark on non-symmetric games:**
>
> > What are the barriers preventing you from extending your results to non-symmetric zero-sum games?
>
> **Response:** We would like to point out that, in fact, the game in question does *not* have to be symmetric. It is sufficient if the min-max value of the game is zero. In this case, all our results hold immediately without modifying the statement or proof. The reason we pointed to symmetric zero-sum games as an important example is merely that, in such games, it is easy to prove that its value is *always* zero. However, there are many asymmetric games in which the value is still zero, and our results readily hold for these games, too. In general, even if the value $V$ of the game is non-zero, our algorithm is guaranteed an expected payoff of at least $V\cdot T - 1$.
>
> More generally, we can readily apply our results even to EFGs that are not zero-sum, or not even two-player (with exactly the same proofs but cost functions defined accordingly). In this case, we guarantee the same expected payoff as the chosen baseline comparator $\mu^c$ (which is considered *safe* in some sense) up to one additive unit, while still having $O(\sqrt{T})$ regret to any strategy $\mu$. Our motivation for stating the results for symmetric zero-sum games was mainly for illustrative purposes.
>
> We will move these remarks to a more prominent place in the writing so it is clear that the limitations are not necessary.
>
> ---
>
> > Or to no-swap-regret?
>
> **Response:** The obvious idea would be to combine the classical reduction of Blum & Mansour (BM) with our phased algorithm. However, this poses several technical challenges, among which are the following: i) It is unclear how to check the regret condition for entering a new phase for swap regret, as there are exponentially many swap functions. ii) If, instead, we check the regret condition for the $A$ different no-regret algorithms in (BM) individually, then these algorithms may be in different phases, and it is unclear how to combine them into a single policy. Hence, whether we can provide the same guarantees for no-swap regret remains an interesting open question.

---

### Official Review · Reviewer_XQ49 · 2025-02-27

**Overall Recommendation:** 3

**Summary:**

In this paper, the goal is to develop a bandit algorithm that guarantees simultaneously constant regret with respect to a given strategy and $\sqrt{T}$ regret with respect to the best strategy in hindsight.

**Claims And Evidence:**

- An extension of the phased algorithm of Even-Dar to the bandit case
- An analysis showing good regret with respect to both the given strategy and the best strategy in hindsight as long as the given strategy has a probability of playing any arm bounded away from zero.
- An application of their approach to NFG and EFG (with lower bounds).

I think the above claims are supported by clear and convincing evidence.

**Essential References Not Discussed:**

I did not find any missing references.

**Experimental Designs Or Analyses:**

I did not check the experimental designs or analyses.

**Methods And Evaluation Criteria:**

The main focus of this work is theoretical. Some simulations are presented based on the game of Kuhn poker. I mostly see them as checks that the theory indeed works.

**Other Comments Or Suggestions:**

See "strengths and weaknesses" section.

**Other Strengths And Weaknesses:**

I think the paper is overall well written and the idea of looking for a strategy that simultaneously guarantees constant regret with respect to another (known strategy) and $\sqrt{T}$ regret with respect to the best strategy in hindsight is interesting.

I think the novelty of the work can be challenged
- The algorithm presented by the authors is really close to Even-Dar et al. (2008). Authors should focus on what are the differences beyond simply using Exp3 as a base algorithm.
- What is the novelty in the presented reduction from OLM to NFG or EFG ?

**Questions For Authors:**

See "strengths and weaknesses" section.

**Relation To Broader Scientific Literature:**

I think the relevant literature is correctly cited.

**Theoretical Claims:**

I did not check the details of the proofs of theoretical claims. The claims themselves seem plausible.

---

> ### Author Rebuttal · Authors · 2025-03-30
>
> Thank you for carefully reviewing our paper and providing valuable feedback!
>
> > The algorithm presented by the authors is really close to Even-Dar et al. (2008). Authors should focus on what are the differences beyond simply using Exp3 as a base algorithm.
>
> **Response:** Indeed, the high-level difference between the algorithms is using Exp3 as a base algorithm. In addition to this:
> - We use importance weighting according to the algorithm's iterates $\mu^t$, not the Exp3 iterates $\hat{\mu}^t$ as usual.
> - We modify the condition for entering a new phase as the true cost vectors are not observed.
> - We switch the learning rate in the last possible phase (see below).
>
> This being said, the main difference lies in the analysis of the algorithm (Section 3.1/Regret Analysis), which is not just a direct combination of the analysis by Even-Dar et al. and the standard Exp3 guarantee. One technical difficulty lies in the fact that the estimated cost functions may be unbounded in general. Our analysis tackles this by a refined treatment of the algorithm's phases: In the last possible phase, the cost estimates can be unbounded but it is sufficient to resort to a regret bound in expectation (which does not need boundedness). In all other phases, for the analysis to go through we need an estimated regret bound with probability one. This is only possible since we are able to bound the estimated costs in these phases due to the comparator assumption and the phasing scheme.
>
> ---
>
> > What is the novelty in the presented reduction from OLM to NFG or EFG ?
>
>
> **Response:** Technically speaking, the application of safe OLM algorithms to NFGs/EFGs by using e.g. the min-max equilibrium as comparator is relatively straightforward (c.f. Section 2). However, we are not aware of any prior work making this interesting connection to learning in games, even under the easier full-information feedback. We thus view it as an important conceptual contribution of our work.
>
> ---
>
>
> **PS:** Please refer to our **Remark on non-symmetric games** in reply to reviewer aZNc for a common response on the generality of our results beyond symmetric games.

---

### Official Review · Reviewer_ZxCd · 2025-03-14

**Overall Recommendation:** 3

**Summary:**

The paper proposes a bandit algorithm for two-player symmetric zero-sum games that guarantees $O(1)$ regret against the minimax strategy and $O(\sqrt{T})$ regret against any strategy.

## update after rebuttal
I keep my score, which remains positive.

**Claims And Evidence:**

I do not see any problematic claims.

**Essential References Not Discussed:**

There does not appear to be missing essential references.

**Experimental Designs Or Analyses:**

The numerical experiments seem sound.

**Methods And Evaluation Criteria:**

I do not see any major issues with the methods.

**Other Comments Or Suggestions:**

Further discussions and comparisons with Lattimore (2015) and Ganzfried & Sandholm (2015) will be appreciated.

**Other Strengths And Weaknesses:**

The regret guarantees in both directions is a strength.

The related work and its comparisons with the manuscript can be more adequately discussed.

**Questions For Authors:**

I have no further questions.

**Relation To Broader Scientific Literature:**

The developments seem novel and of interest to a variety of fields because of the game theoretic implications.

**Theoretical Claims:**

The proofs seem correct.

---

> ### Author Rebuttal · Authors · 2025-03-30
>
> Thank you for carefully reviewing our paper and providing valuable feedback!
>
> > The related work and its comparisons with the manuscript can be more adequately discussed.
>
> **Response:** We agree that further elaboration on the related work would increase the clarity of the paper. In the current version, an extensive discussion of related work is deferred to Appendix B. We plan to move parts of this to the main text in the camera-ready version.
>
> ---
>
> > Further discussions and comparisons with Lattimore (2015) and Ganzfried & Sandholm (2015) will be appreciated.
>
> **Response:** Further comparisons with Lattimore (2015):
>
> Lattimore (2015) shows that in multi-armed bandits, $O(1)$ regret compared to a single comparator action implies a worst-case regret of $\Omega(AT)$ compared to some other action. In our terminology, a single action corresponds to a deterministic strategy. We show that, perhaps surprisingly, it is possible to circumvent this lower bound if the comparator strategy plays each action with some non-zero probability $\delta>0$ (while maintaining the optimal order of $\sqrt{T}$ regret). As further discussed in the main text, this minimal possible assumption is sensible in various game-theoretic contexts.
>
>
> Additional comments regarding Ganzfried & Sandholm (2015):
>
> Ganzfried & Sandholm (2015) do not provide any sort of regret guarantees. Instead, they ask the following question: In which rounds of the game is it possible to deviate from the min-max strategy? Their algorithmic approaches rely on best-responding to an *opponent model* whenever the algorithm has accumulated just enough utility to risk losing it again. While the authors do prove safety guarantees, this is not the case for *exploitation*. The latter would likely heavily depend on the other player and the quality of the opponent modeling. In our paper, we do not encounter this difficulty because we consider standard regret minimization against any (time-varying) adversary.
>
> ---
>
>
> **PS:** Please refer to our **Remark on non-symmetric games** in reply to reviewer aZNc for a common response on the generality of our results beyond symmetric games.

---

> > ### Comment · Reviewer_ZxCd · 2025-04-07
> >
> > Thank you for the response. I have no further questions. My score remains positive.

---

### Official Review · Reviewer_JKhZ · 2025-03-14

**Overall Recommendation:** 3

**Summary:**

This paper derives regret upper bounds that vary depending on the comparator in the setting of online learning with bandit feedback. Specifically, the paper proposes an algorithm that simultaneously achieves an $O(1)$ regret upper bound when the comparator of the regret lies in the interior of the probability simplex, which is the feasible region, and an $O(\sqrt{T})$ regret upper bound otherwise.
The authors primarily discuss this in the context of symmetric two-player zero-sum normal-form games and further extend their analysis to extensive-form games.

**Claims And Evidence:**

Yes, all propositions and claims in the paper are with proofs or references.

**Essential References Not Discussed:**

no

**Experimental Designs Or Analyses:**

Yes, I have briefly checked the experimental setup and results and confirmed that there are no major issues.

**Methods And Evaluation Criteria:**

Yes, the proposed algorithms are variants of existing algorithms in online learning and are thus valid. Moreover, the evaluation metrics (i.e. regrets) are standard in the literature.

**Other Comments Or Suggestions:**

no

**Other Strengths And Weaknesses:**

The major issue with this paper is the unclear motivation for its contributions. The authors derive different regret upper bounds depending on whether the comparator, but it is difficult to see how these bounds offer advantages from the perspective of learning in games. What are the benefits of achieving a lower regret when choosing the minimax policy ${argmin}_\mu \max_{\nu} V(\mu, \nu)$ as the comparator? If there are advantages to using the minimax policy as the comparator in the context of learning in games, the reviewer would like the authors to clarify them.

Additionally, some statements are inaccurate, and certain terms are used in a non-standard manner, making the paper a little harder to understand. For example, in Line 17, the statement "the regret compared to any fixed strategy $\mu$ satisfies … $\leq O(\sqrt{T})$" depends on the algorithm being used, and such a regret upper bound is not necessarily always achievable. Moreover, in Line 46, the term "the best strategy" is ambiguous, making it difficult to understand. Similarly, in Line 89, the phrase "a special ('safe') comparator strategy" is unclear at this point. Additionally, in Line 104, the term  "the worst-case expected regret $\max_{\mu} \mathcal{R}(\mu)$" is used, but referring to the worst case over the comparator (rather than over a sequence of gradients) in this way is maybe uncommon.

This work focuses on symmetric games. Can the authors address this limitation?

It is possible that I have not fully understood the motivation behind this work. If I receive a convincing response on this point, I am willing to raise my score.

**Questions For Authors:**

The reviewer expects the authors to address the questions raised in Other Strengths and Weaknesses.

**Relation To Broader Scientific Literature:**

In the context of games, I have not seen an approach that achieves different regret upper bounds depending on the comparator.
From the paper, the motivation of deriving different regret upper bounds based on the comparator is unclear.
For more detailed comments, see the section Other Strengths and Weaknesses below.

**Theoretical Claims:**

I have reviewed the claims in the main body and confirmed that there are no major issues.

---

> ### Author Rebuttal · Authors · 2025-03-30
>
> Thank you for carefully reviewing our paper and providing valuable feedback!
>
>
> > It is possible that I have not fully understood the motivation behind this work. If I receive a convincing response on this point, I am willing to raise my score.
>
> **Response:** From the viewpoint of learning in games, the motivation of our paper is the following: Suppose you play a repeated game such as Rock-Paper-Scissors or Heads-Up Poker against an unknown opponent. Should you rather run a no-regret algorithm, or simply play the Nash equilibrium? The former would guarantee that you linearly exploit the opponent when, for example, they decide to play a static policy that is slightly suboptimal (non-equilibrium). The latter would guarantee you to not lose money (beyond the Nash value of $0$) in expectation. But it is easy to see that neither guarantees *both* of these preferable properties. Our work states that you can achieve essentially both simultaneously under the given minimal assumption.
>
>
> ---
>
>
> > What are the benefits of achieving a lower regret when choosing the minimax policy $\arg\min_{\mu} \max_{\nu} V(\mu, \nu)$ as the comparator? If there are advantages to using the minimax policy as the comparator in the context of learning in games, the reviewer would like the authors to clarify them.
>
> **Response:** This is indeed a key motivation behind our work (c.f. Section 2): Notice that for the minimax policy $\mu^\star$, we have $V(\mu^\star,\nu^t) \leq 0$ no matter what policy $\nu^t$ Bob plays. Hence, a regret bound of $\mathcal{R}(\mu^\star) \leq 1$ compared to the minimax policy implies that $\sum_{t=1}^T \mathbb{E}[V(\mu^t,\nu^t)] - 0\cdot T \leq 1$. This proves that we lose at most $1$ unit to Bob throughout the interactive play (in expectation), no matter Bob's play. Simply running a standard no-regret algorithm would not guarantee this; it can lose up to $\sqrt{T}$ units, which can be a significant amount. But, since we simultaneously maintain $O(\sqrt{T})$ regret compared to *any* $\mu\neq\mu^\star$, we still enjoy all the benefits of playing a no-regret algorithm, including that we linearly exploit, for example, a static opponent that slightly deviates from the equilibrium.
>
>
> ---
>
>
> > This work focuses on symmetric games. Can the authors address this limitation?
>
>
> **Response:** Yes! Our results hold beyond symmetric games, which only serve as an important motivation. Please refer to our **Remark on non-symmetric games** in reply to reviewer aZNc.
>
>
>
>
> ---
> ---
>
>
> **Additional minor clarifications:**
>
> > In Line 17, the statement "the regret compared to any fixed strategy $\mu$ satisfies $O(\sqrt{T})$" depends on the algorithm being used, and such a regret upper bound is not necessarily always achievable.
>
> **Response:** Here, we are referring to any no-regret algorithm that has regret of $O(\sqrt{T})$ uniformly over adversaries and comparators. Such no-regret algorithms exist both for normal- and extensive-form games (for example online mirror descent). The property itself does not depend on the specifics of the algorithm and holds for any such no-regret algorithm.
>
> > In Line 46, the term "the best strategy" is ambiguous, making it difficult to understand.
>
> **Response:** By regret compared to the best strategy in hindsight, we mean that the regret bound holds compared to any strategy $\mu$ (in particular to a minimizer of the observed sequence of play). We are happy to change the wording to "against any fixed strategy $\mu$" for clarity.
>
> > in Line 89, the phrase "a special ('safe') comparator strategy" is unclear at this point.
>
> **Response:** We will change this to the following in the final version: *Alice receives a special comparator $\mu^c$ that she considers a 'safe' strategy. The motivation for this is that we can later choose $\mu^c$ to be a minimax equilibrium $\mu^\star$, which is safe in the sense of guaranteeing zero expected loss in symmetric zero-sum games.*
>
> > In Line 104, the term "the worst-case expected regret $\max_{\mu} \mathcal{R}(\mu)$" is used, but referring to the worst case over the comparator (rather than over a sequence of gradients) in this way is maybe uncommon.
>
> **Response:** We agree that here the regret refers to the worst case over the comparator. (Note that since our bounds are uniform over the opponent's play, they also hold in the worst case over the sequence of gradients.) We are willing to change this to simply "regret" (as opposed to "regret compared to $\mu$") throughout to avoid confusion.

---

> > ### Comment · Reviewer_JKhZ · 2025-04-08
> >
> > Thank you for the detailed response.
> > Since my concerns regarding the motivation have been resolved, I have raised my score from 2 to 3.
> >
> > That said, there still appear to be several issues.
> > First, there seems to be significant room for improvement in the writing and presentation.
> > Additionally, as pointed out by other reviewers, the case of non-symmetric games is not sufficiently discussed.
> > Lastly, references that support the motivation seem to be not discussed (the reviewer guesses that there may be existing literature that discusses similar motivations).
> >
> > It is strongly expected that these issues will be addressed in the revised version.

---

> > > ### Author Response · Authors · 2025-04-09
> > >
> > > Thank you for your response and the positive evaluation. We greatly appreciate the additional suggestions and will carefully incorporate them into the camera-ready version. We believe that the extra page will leave ample space for this.

---

### Decision · Program_Chairs · 2025-05-01

**Decision:**

Accept (poster)

**Comment:**

This paper studies the following "best of both worlds" problem: in a minimax game, a player would like to simultaneously achieve a constant regret against the Nash equilibrium strategy, and a $\widetilde{O}(\sqrt{T})$ regret against all fixed plays. Although in general this task was shown to be impossible in the literature, for symmetric zero-sum games with a lower bound on the pmf of the mixed Nash equilibrium, this task is shown to be possible for both normal and extensive forms. A tight lower bound on the pmf lower bound is also given in this paper.

All reviewers are positive about this paper, but the support is not super enthusiastic mainly because the lower bound assumption may not be practical in many scenarios. In addition, the algorithmic novelty compared with the existing work is limited. Despite these concerns, reviewers think this paper establishes a significant result, which they (and I) recommend to accept.